

# Unique Microphysical Properties of Small Boundary Layer Ice Particles under Pristine Conditions on Dome C, Antarctica

Adrian Hamel[1], Massimo del Guasta[2], Carl Schmitt[3], Christophe Genthon[4], Emma Järvinen[5], and Martin Schnaiter[5,6]

[1]Institute of Meteorology and Climate Research Atmospheric Aerosol Research (IMKAAF), Karlsruhe Institute of Technology, Karlsruhe, Germany
[2]Istituto Nazionale Ottica CNR, Sesto Fiorentino, 50019 Firenze, Italy
[3]University of Alaska Fairbanks, Fairbanks, USA
[4]Laboratoire de Météorologie Dynamique, IPSL, CNRS, Ecole Normale Supérieure, Sorbonne Université, PSL Research, Paris, France
[5]Institute for Atmospheric and Environmental Research, University of Wuppertal, Wuppertal, Germany
[6]schnaiTEC GmbH, Wuppertal, Germany

**Correspondence:** Adrian Hamel (adrian.hamel@kit.edu)

**Abstract.** The Antarctic plateau, one of the coldest and cleanest regions of our planet, experiences almost exclusively frozen precipitation. Understanding the microphysical properties of inland Antarctic boundary layer ice particles with sizes below a few hundred micrometers is essential to improve atmospheric models and accurately validate remote sensing data for this region. Currently, only a small number of in situ atmospheric measurements exist for particle sizes smaller than $100\,\mu m$

on the Antarctic plateau, performed over short measurement times. We present the first multi-week study of optical in situ measurements of boundary layer ice particle size, shape and morphological complexity for sizes down to $11\,\mu m$ with a temporal resolution in the order of minutes, including a multi-day ice fog event. Classifying cirrus ice fog events with a lidar system, we found mean particle sizes smaller than $11\,\mu m$ for ice fog events and of about $70\,\mu m$ for cirrus precipitation and diamond dust events. The mean particle concentration of the ice fog at Dome C ($3.9\,L^{-1}$) is found to be lower than commonly used

parametrisations of Arctic ice fog and lower than the concentration of anthropogenically influenced urban ice fog measured at Fairbanks, Alaska during a three-year study with the same instrument ($350\,L^{-1}$). Moreover, ice fog particles at Dome C are found to be more pristine than at Fairbanks. Therefore, Antarctic boundary layer ice particles need to be parametrised differently than their Arctic counterparts due to distinct conditions on the Antarctic plateau.

## 1  Introduction

The Antarctic plateau is an extremely cold region exceeding $3000\,m$ above sea level, where essentially all precipitation is frozen with a fraction of about $40\,\%$ clear-sky precipitation (Dittmann et al., 2016). Ice crystals in this cold and dry place regularly have sizes in the order of $10\,\mu m$ and smaller (Walden et al., 2003). In situ measurements of these small atmospheric ice particles remain scarce due to challenging conditions in this remote region, where access is only possible in the few summer months of the year. At the same time Antarctica is one of the last regions on our planet with very small impact from human





activity. It enables us to study cirrus formation in an environment that can be considered one of the cleanest on the planet (Heumann, 1993). While there have been several measurement campaigns with detailed measurements of Antarctic boundary layer ice particles using optical array probes (Lawson et al., 2006; Lawson and Gettelman, 2014), imaging of precipitation on flat surface benches (Walden et al., 2003; Schlosser et al., 2016) and a flatbed scanner (Del Guasta, 2022) they cannot reliably determine ice crystal shape, complexity and size distribution for particles with sizes below $50\,\mu m$ (Kaye et al., 2008;

Ulanowski et al., 2010). Scanning electron microscopy of Formvar replica can obtain shape and size information in this size range. Nevertheless, it is currently only available for some tens of snap-shots covering a measurement time of $30\,s$ each (Santachiara et al., 2016). There is insufficient information about ice particle morphology for ice particles with sizes below $50\,\mu m$ (Gultepe et al., 2017). This information is needed to accurately determine the cloud radiative impact of the boundary layer ice crystals that are abundant in polar regions.

Small atmospheric ice particles at ground level are commonly classified as ice fog or diamond dust. Ice fog is a fog composed of suspended ice particles reducing visibility (American Meteorological Society, b; World Meteorological Organization, b). Diamond dust is defined as ice particles falling from an apparently cloudless sky (American Meteorological Society, a; World Meteorological Organization, a). In a commonly used parametrisation ice fog particles have sizes below $30\,\mu m$ with concentrations above $1\cdot10^3\,L^{-1}$ and diamond dust particles have sizes larger than $30\,\mu m$ with concentrations below $4\cdot10^3\,L^{-1}$

(Girard and Blanchet, 2001a). However, reported sizes of ice fog tend to fall outside these limits and the separation is not always clear (Gultepe et al., 2017). During winter time falling diamond dust is simulated to increase the downward infrared radiative flux by up to $60\,W/m^2$ and ice fog of about $7.4\,W/m^2$ leading to a surface warming rate of $2.85\,K/day$ and $2.57\,K/day$, respectively (Girard and Blanchet, 2001b).

Vignon et al. (2022) has investigated locally formed ice fog events at Dome C, Antarctica that do not have a liquid origin

suggesting that the two observed ice fog events are initiated by homogeneous freezing of solution aerosol particles. However, they have not found a direct proof missing measurements of ice particle microphysics. The single particle counter and diffraction pattern imager PPD-2K (Kaye et al., 2008) can add important information to the open question of the particle morphology, radiative properties and the nucleation process in locally formed Antarctic ice fog. Its capability to measure ice crystal size, complexity and shape information in ice fog down to particle sizes of a few microns has been reported in previous studies

of boundary layer ice crystal in an urban environment in Fairbanks, Alaska (Schmitt et al., 2024a, b), and in cloud chamber simulation experiments (Schnaiter et al., 2016). Furthermore, the in situ measurements of microphysical properties of low-level atmospheric ice crystals are valuable for the validation of satellite data retrievals and atmospheric modelling of inland Antarctica that currently depend on a small number of measurements (Palm et al., 2011).

During our field campaign the PPD-2K was deployed on the roof of the physics shelter at Concordia station from 21 Novem-

ber 2023 to 5 January 2024. The Italian and French Antarctic base is located at Dome C at 75°S 123°E on the Antarctic plateau at an altitude of 3233 m above sea level. The instrument measured over 700 000 ice particles with spherical equivalent diameters between $11\,\mu m$ and $150\,\mu m$. The temporal evolution of exemplary ice fog and cirrus precipitation events are shown in combination with atmospheric measurements, including temperature and relative humidity at different altitudes, wind speed and direction and lidar backscattering signal and linear depolarisation. The microphysical properties of the boundary layer ice





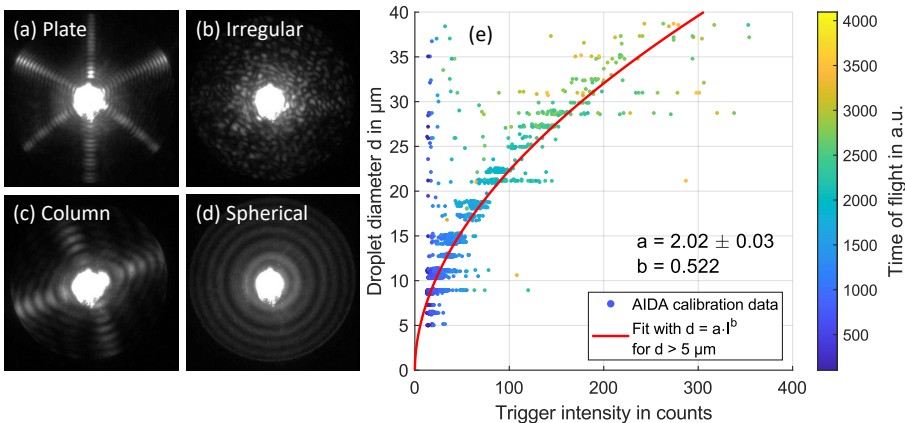

**Figure 1.** Example diffraction patterns for a plate-like **(a)**, an irregular **(b)**, a columnar **(c)** and a spherical particle **(d)** can be seen. **(e)** shows the calibration dataset of droplet diameters derived from the Mie fringes plotted over the trigger intensity measured with PPD-2K. The fit that is used for the size conversion of the trigger intensity measurements is added.

crystals in this clean region are shown separately for lidar classified ice fog events and events consisting of diamond dust or thin cirrus precipitation. The results are compared to a three-year study of ice fog in Fairbanks, Alaska between 2020 and 2022 in an urban environment with strong anthropogenic influences, such as the emission of water vapour and combustion particles into the atmosphere (Schmitt et al., 2013, 2024b). This provides new insights on the microphysical properties of boundary layer ice particles smaller than 100 μm on the Antarctic plateau.

# 2 Methods

## 2.1 Particle Phase Discriminator 2 Karlsruhe edition (PPD-2K)

The Particle Phase Discriminator 2 Karlsruhe edition (PPD-2K) is a single particle counter that measures the forward scattered light for particles with sizes from about 11 μm to 150 μm and takes images of the diffraction patterns in an annulus between 7.4° and 25.6° (Kaye et al., 2008). It was developed at the University of Hertfordshire and is the laboratory version of the
Small Ice Detector-3 (SID-3) (Ulanowski et al., 2014). A 100 mW frequency-doubled Nd:Yag laser emitting at a wavelength of 532 nm is used as light source. The sensitive area of the instrument is defined by an overlap of the laser beam and the trigger field of view. When a particle passes the sensitive area 8 % of the forward scattered light is reflected to a photomultiplier tube (PMT) using a beam splitting mirror. The recorded intensity of the single particles is used to determine particle size and concentration with a maximum count rate of 11 kHz. The majority (92 %) of the forward scattered light is directed to
an intensified Photek ICCD218 camera with a resolution of 582 × 592 pixels. It takes images of the diffraction patterns of a subsample of the particles with a maximum repetition rate of 30 Hz. For ice crystals the forward scattering images show diffraction patterns, which can be used to get information about the particle shapes (Kaye et al., 2008; Vochezer et al., 2016)



and the particle small-scale complexity (Ulanowski et al., 2010; Schnaiter et al., 2016). In this work the forward scattering patterns are analysed with the Fourier method by Vochezer et al. (2016). A Fourier fit is applied to the polar integrated intensity
profile. Maximum Fourier coefficients of 2 and 4 identify diffraction patterns from columnar hexagonal ice particles (see Fig. 1c) and maximum Fourier coefficients of 3 and 6 identify plate-like columnar ice particles (see Fig. 1a). Spherical particles can be identified by characteristic Mie fringes in polar direction (see Fig. 1d). All other particles are classified as irregular (see Fig. 1b). It should be noted that sufficiently rough hexagonal particles may show irregular diffraction patterns and can thus be classified as irregular. Therefore, the fraction of plates and columns needs to be seen as a lower bound. Furthermore, a machine
learning convolutional neural network method has been introduced by Schmitt et al. (2024b) to analyse the PPD-2K scattering patterns. The neural network was trained on hand-selected particles and can identify additional habits such as sublimating, pristine and rough ice crystals.

Besides information about the particle shape also particle complexity can be derived from the diffraction patterns. For this the gray-level co-occurrence matrix (GLCM) method is used (Lu et al., 2006). The method has been applied to ice crystal
diffraction patterns by Ulanowski et al. (2010, 2014) and Schnaiter et al. (2016). The GLCM describes how often pairs of gray-level occur for pixels separated by a distance $\Delta d$ along a defined direction. Schnaiter et al. (2016) found the normalised energy feature parameter $k_e$ as the most robust parameter to measure small-scale optical complexity from the diffraction patterns. $k_e$ is the coefficient of an exponential fit to the sum of the squared elements of the GLCM for changing pixel distance $\Delta d$. It is calculated for diffraction patterns with mean intensities between 10 and 25 counts to reduce the bias caused by intensity
variations of the diffraction pattern images. The parameter varies between 3.8 and 6.0, and a threshold for rough ice crystals of $k_e^{thr} = 4.6$ has been defined by Schnaiter et al. (2016) for measurements with the SID-3 instrument, the airborne version of the PPD-2K. An overview of PPD-2K diffraction patterns of columnar particles from Dome C with a range of different $k_e$ values is shown in appendix B. Speckles outside the 22° halo angle progressively appear on the diffraction patterns for $k_e$ values of about 4.6 and higher for the laboratory instrument PPD-2K, similar to the airborne version SID-3. Therefore, the threshold
value of $k_e$ for discriminating optically rough from optically smooth ice crystals for PPD-2K is similar to the SID-3 threshold of 4.6. In this work, spherical particles are not considered for the calculation of the mean of $k_e$ because the diffraction patterns with Mie fringes would falsely increase the mean complexity.

The instrument has been calibrated at the AIDA cloud chamber of the Karlsruhe Institute of Technology in May 2023 for the Dome C deployment. The calibration followed the procedure of Vochezer et al. (2016). Data from multiple spray cloud
experiments are used, where droplets in a variety of sizes are created in the AIDA cloud chamber. The size of the droplets can be determined from the number and positions of Mie fringes that are seen on the diffraction pattern images. In Fig. 1e the size that is derived from the Mie fringes is shown as a function of the trigger intensity. A function of the form

$$d = a \cdot I^b \tag{1}$$

is fitted to the data, where $I$ is the trigger intensity, $b = 0.522$ is a factor that depends on the PPD-2K trigger geometry and $d$ is
the spherical equivalent diameter. This function is used to determine the particle size in this study. The measurement uncertainty in size is 1.5 % based on the covariance matrix of the function fit. The spherical equivalent diameter is the diameter of a droplet



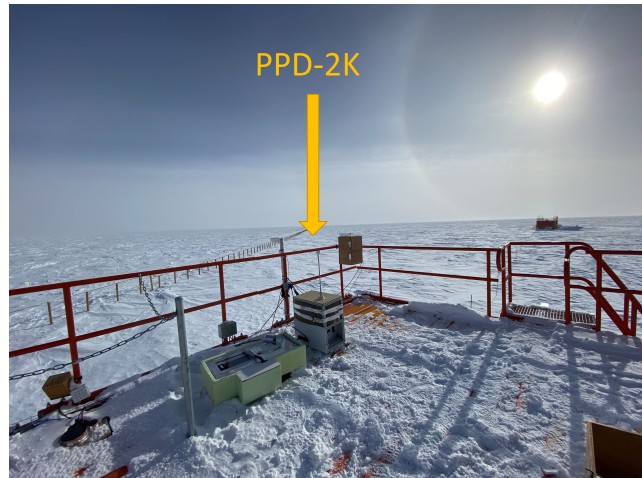

**Figure 2.** PPD-2K single particle counter and diffraction pattern imager at its measurement site on the roof of the physics shelter at Concordia station, Antarctica. In the background a 22° halo around the sun appears from the light scattered by small hexagonal ice particles in the atmosphere (diamond dust).

that scatters the same intensity in the direction of the trigger field of view as the recorded particle. The maximum dimension of the ice particle is estimated to be 1 to 2.5 times larger than the spherical equivalent diameter depending on the particle size, shape and complexity. The estimation is based on raytracing simulations of the light that is scattered in the direction of the PPD-

2K trigger field of view by ice particles of different sizes and shapes (Yang et al., 2013). At the measurement site at Dome C, the air flow is produced by a pump set to a mass flow of $(5.0 \pm 0.5)$ normal liters per minute. The relative uncertainty in particle concentration is 10 % due to the uncertainty in sample flow. After every particle event the PPD-2K has an electronic dead time of $8.25\,\mu s$, which leads to a reduced sample volume. This reduced sample volume is taken into account in the calculation of the particle concentration similar to Vochezer et al. (2016). The maximum applied concentration correction is less than 1.5 %

during the deployment at Dome C and Fairbanks with maximum detected particle concentrations of $2.1 \cdot 10^4\,L^{-1}$. The particle concentration of PPD-2K is the concentration of particles larger than the minimum detection size of $11\,\mu m$.

At Dome C the instrument sometimes recorded bursts of triggers on noise with constant low intensity at a count rate orders of magnitude larger than the operational measurements. This happened especially during the start-up phase of the instrument. These short periods with only empty images are not considered in the analysis. This is done with the following procedure.

First, particle counts are integrated over $10\,s$ intervals. Time intervals containing multiple consecutive empty images (mean intensity below the upper background level) are excluded. Then, the average concentration of $600\,s$ time periods is calculated only using the valid $10\,s$ time bins. The $600\,s$ averaging period ensures robust counting statistics at low particle concentrations. It is comparable to the averaging periods of ice particle concentration measurements at South Pole station by Lawson et al. (2006) and slightly higher than for ice fog measurements at Fairbanks, Alaska, where higher mean particle concentrations

occurred (Schmitt et al., 2024b).



**Table 1.** This table shows the wind speed and wind direction limits when PPD-2K data are excluded due to the possibility of pollution from the exhausts of the generators of Concordia station. The wind direction and speed limits are based on aerosol measurements of Virkkula et al. (2022)

| Wind speed $v$ in m/s | Lower excluded wind direction in ° | Upper excluded wind direction in ° |
|:---:|:---:|:---:|
| $v>2$ | 20 | 110 |
| $1<v<2$ | 0 | 150 |
| $v<1$ | 0 | 360 |

The measurement site for the deployment was the roof of the physics shelter of the Italian and French Antarctic base Concordia station at Dome C at 75°S 123°E. This is the top of an ice dome on the Antarctic plateau at an altitude of 3233 m above sea level. The surrounding area is of flat topography with no noticeable hills or valleys. During the measurement period from 21 November 2023 to 5 January 2024 the air temperature ranged between -45 °C and -22 °C with a mean air pressure

of (646±6) hPa according to the automated weather station operated at this location (Grigioni et al., 2022). The instrument was under remote operation with at least daily access during the first three weeks and no access for the last four weeks of the deployment. The inlet is situated approximately 6 m above ground level with an upward facing horn shaped nozzle (diameter: 39 mm) to minimize sampling artefacts. The connection to the PPD-2K was a metal tube in a vertical straight line and a mass flow of (5.0 ± 0.5) normal liters per minute. The instrument was placed outside and operated in intervals of 170 min with

pauses of 10 min in between. Occasional downtimes occurred during the no access period due to connection errors between the computer and the instrument. The operational times are shown in Appendix D. The physics shelter is located about 500 meters south of Concordia station, upstream with regard to the predominant wind direction, in a clean zone, where no motorized vehicle operation is allowed in order to minimize the effects of aerosol pollution and water vapour emission by combustion exhausts. PPD-2K measurements are excluded for the times when the wind direction did not rule out pollution from the

generators of Concordia station. The excluded wind directions are based on aerosol measurements of Virkkula et al. (2022) and can be seen in table 1. For wind speeds lower than 1 m/s the data are always excluded because aerosol pollution is possible for any wind direction. For wind speeds 1 m/s<$v$<2 m/s wind directions between 0° and 150° are excluded and for wind speed $v$>2 m/s wind directions between 20° and 110° are excluded. For high wind speeds drifting snow can occur on the Antarctic plateau. To have an increased particle count rate at the measurement height of PPD-2K the wind speed must exceed about 7 m/s

as seen in the multi-year results of the flatbed scanner by Del Guasta (2022).

## 2.2 Atmospheric instrumentation

The Consiglio Nazionale delle Ricerche - Istituto Nazionale di Ottica (CNR-INO) depolarisation lidar has been operated year-around since 2008 at the physics shelter of Concordia station. The system measures the backscattered intensity and the linear depolarisation ratio of aerosol and clouds in the troposphere between 10 m and 12 km above ground (Del Guasta et al., 2024).

It uses a linearly polarised laser with a wavelength of 532 nm and produces profiles with vertical resolution of 7.5 m and a





temporal resolution of 5 min. The output data are the lidar backscattering signal and the linear depolarisation ratio $\delta$:

$$\delta = \frac{I_{\mathrm{PER}}}{I_{\mathrm{PAR}}} \tag{2}$$

with the intensity of the backscattered light with polarisation perpendicular ($I_{\mathrm{PER}}$) and parallel ($I_{\mathrm{PAR}}$) to the polarisation of the emitted light. The lidar signal is proportional to the particle concentration and particle scattering cross-section. The linear

depolarisation ratio $\delta$ gives information about the particle shape and, consequently, phase. Spherical (liquid) particles result in $\delta$ close to zero, whereas non-spherical particles show a non-zero linear depolarisation ratio (Wang and Sassen, 2001).

The ground temperature, wind speed and wind direction at ground level are recorded with an automated weather station (AWS) (Grigioni et al., 2022). Temperature and relative humidity are also recorded at different altitude levels between 3 m and 42 m above ground on a tower located at Concordia station (Genthon et al., 2022). The measurements are taken with Vaisala

HMP155 humidity and temperature probes with heated inlets. The relative humidity with respect to ice is calculated from the relative humidity of the heated air, the temperature of the heated air and the temperature of the ambient air. This method avoids inaccurate measurement due to ice deposition on the sensor in an environment, where high supersaturation with respect to ice regularly occurs. The parametrisation of Murphy and Koop (2005) is used for the calculation. The measurement uncertainty at air temperature above -45 °C is 7 % for the relative humidity with respect to ice, 4.5 % for the relative humidity with respect to

water and 0.4 K for the ambient temperature (Vignon et al., 2022).

Air-parcel back-trajectories are calculated with the National Oceanic and Atmospheric Administration (NOAA) Hybrid Single-Particle Lagrangian Integrated Trajectory model (HYSPLIT) using the Global Forecast System (GFS) with a grid size of 0.25° x 0.25° (see https://www.ready.noaa.gov/HYSPLIT.php, last accessed on 22 May 2025) (Rolph et al., 2017; Stein et al., 2015).

## 3    Results

First, an overview of the particle habits under different atmospheric conditions is given. Then, a three-day temporal evolution of an exemplary ice fog event is presented. The ice fog event motivates a lidar-based classification of ice fog periods in comparison to cirrus precipitation and diamond dust periods. Finally, the microphysical properties of the different events of boundary layer ice particles are analysed and compared to ice fog data from Fairbanks.

### 3.1    Habit fractions for different atmospheric conditions

PPD-2K had an uptime of 57.8 % during the 47-day measurement period. For 29.3 % of the uptime, wind speeds and directions were observed within the contamination thresholds given in Tab. 1. These measurement times were excluded from the data analysis due to possible aerosol pollution from Concordia station. This gives a data coverage of 40.8 % during the measurement period. Fig. 3 shows the habit fractions determined with the machine learning convolutional neural network analysis of

the diffraction patterns (Schmitt et al., 2024a). The fractions are shown separately for different temperature ranges, relative humidity, temperature inversion and particle size. In Fig. 3a it can be seen that the fraction of rough particles was with 21 %



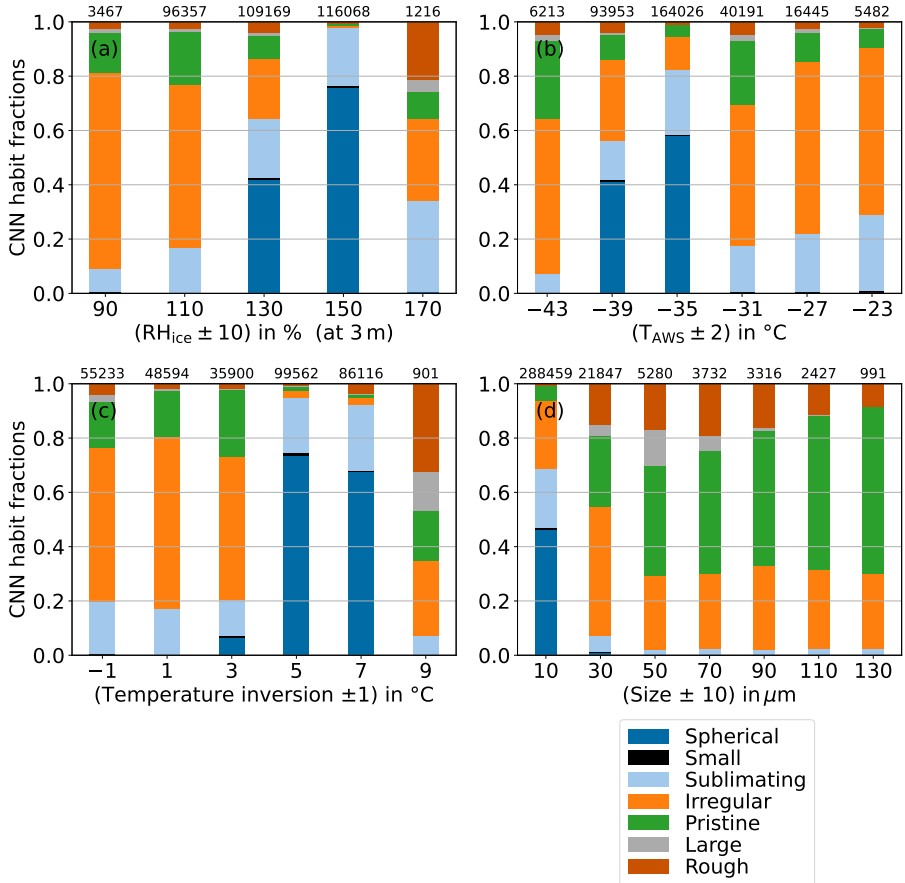

**Figure 3.** Particle fractions determined with the machine learning convolutional neural network method (Schmitt et al., 2024b) for different ranges of relative humidity with respect to ice ($RH_{ice}$) **(a)**, automated weather station temperature ($T_{AWS}$) **(b)**, temperature inversion **(c)** and particle size **(d)**. The temperature inversion is defined as the difference in air temperature measured 42 m above ground and 3 m above ground at the meteorological tower of Concordia station. Only non-pollution times are included in the analysis.

the highest for relative humidity with respect to ice between 160 % and 180 %. This is expected from a more complex growth regime at high relative humidity (Bailey and Hallett, 2009; Schnaiter et al., 2016). In all other relative humidity ranges the fraction of rough particles was below 5 %. Furthermore, the largest fraction of spherical particles (droplets) was recorded for

a relative humidity with respect to ice between 140 % and 160 %, which is the relative humidity with respect to ice at water saturation at about -35 °C (Koop et al., 2000), where most spherical particles occurred. Surprisingly, the fraction of particles classified as sublimating increased with increasing relative humidity. This is unexpected because sublimating particle shapes are expected below or close to saturation with respect to ice (Schnaiter et al., 2016; Schmitt et al., 2024b). One possible reason is that not only sublimating particles show this type of diffraction pattern. For example, frozen droplets have been observed to

show similar diffraction patterns (Järvinen et al., 2016). Nonetheless, it needs to be considered that the particle growth con-



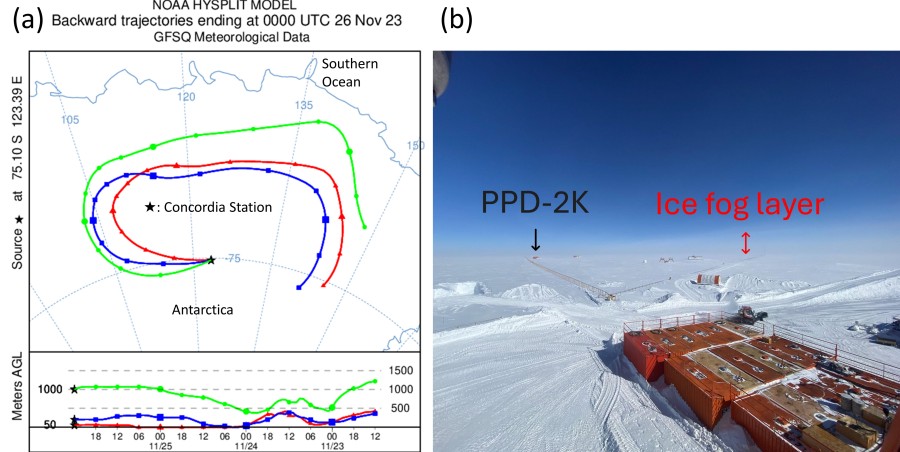

**Figure 4. (a)**: Air parcel back-trajectories at heights of 50 m, 200 m and 1000 m above ground are shown over 84 h, confirming that the ice fog event of 25 November 2023 is locally formed on the Antarctic plateau and does not originate from maritime air masses that are transported inland. The back-trajectories are calculated with the NOAA HYSPLIT model using the Global Forecast System (GFS) with a grid size of 0.25° x 0.25° (see https://www.ready.noaa.gov/HYSPLIT.php, last accessed on 24.01.2024) (Rolph et al., 2017; Stein et al., 2015). **(b)**: Photo from the main building of Concordia station showing the PPD-2K measurement site during an ice fog event. The photo was taken at 00:19 UTC on 26 November 2023 when the ice fog layer was increasing in depth.

ditions could have been different from the relative humidity at the measurement site when the ice particle was transported or sedimented there.

In Fig. 3b it can be noted that the fraction of pristine ice particles decreased and the fraction of sublimating particles increased for increasing temperature. This is in good agreement with previous findings that ice crystals tend to grow to more pristine shapes at colder temperatures (Bailey and Hallett, 2009; Schnaiter et al., 2016). A larger fraction of spherical particles was only found between -40 °C and -34 °C. This was the temperature range of two high concentration supercooled liquid fog events that occurred on 17 and 18 December 2023 and dominated the total recorded particles at these temperature ranges.

Fig. 3c shows the particle shapes depending on the difference between the air temperature measured at the meteorological tower at a height of 42 m above ground and 3 m above ground. Hereafter, the difference between the temperatures at the two levels is referred to as temperature inversion. The trend looks similar to Fig. 3a, which can be explained by high relative humidity often occurring at strong temperature inversions.

Fig. 3d shows the particle fractions depending on particle size. It can be noted that spherical particles almost only occurred at sizes below 20 μm and sublimating particles only occurred at sized below 40 μm. The fraction of pristine particles increased with increasing particle size from about 5 % at particle sizes of about 10 μm to 56 % at particle sizes of about 130 μm.



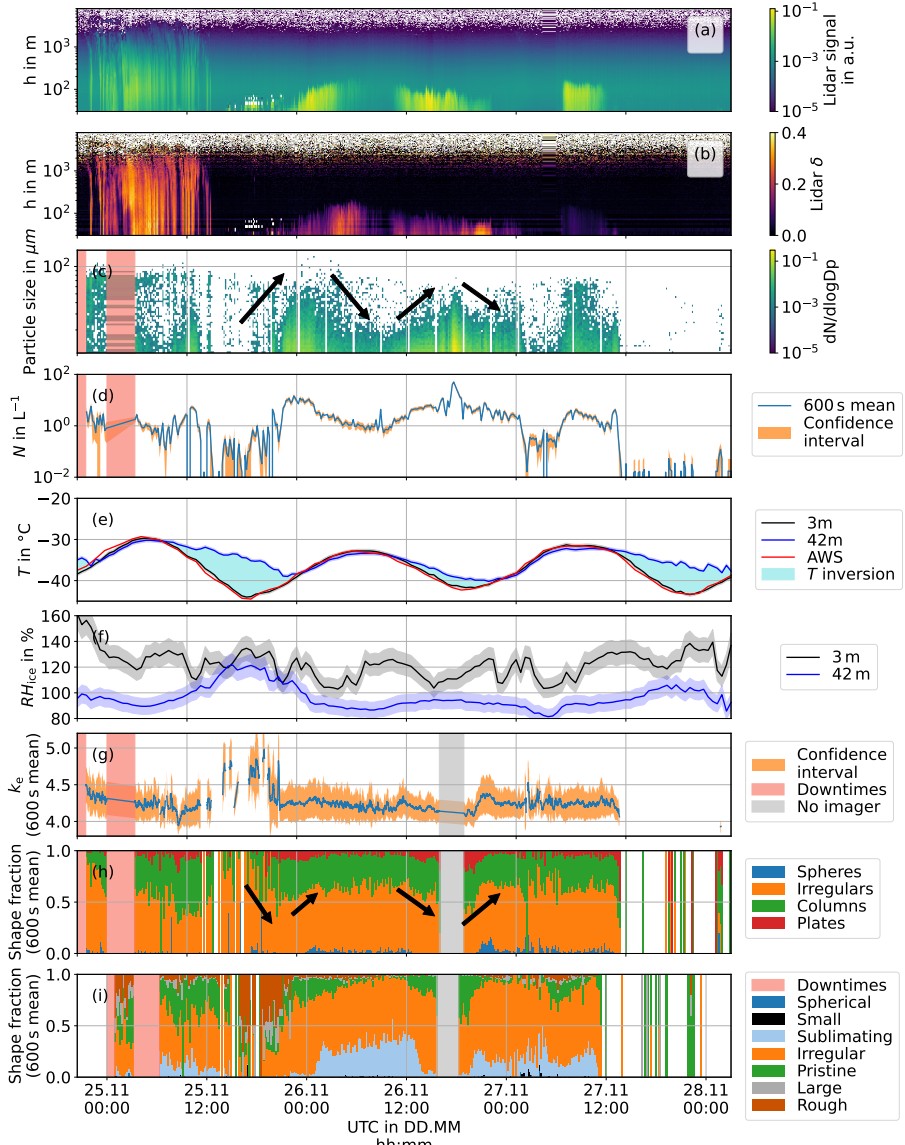

**Figure 5.** Temporal evolution of the ice fog event of 25 November 2023 with lidar backscattering signal in a.u. for the height $h$ in m **(a)**, lidar backscattering linear depolarisation $\delta$ for the height $h$ in m **(b)**, particle size distribution **(c)** and mean particle concentration ($N$) over 600 s **(d)**. Instrument downtimes are shaded in red. **(e)** shows the temperature ($T$) measured at heights of 3 m and 42 m above ground is shown in °C. Temperature inversions are highlighted. **(f)** shows the relative humidity with respect to ice ($RH_{ice}$) at heights of 3 m and 42 m. **(g)** shows the 600 s rolling mean ice particle small-scale optical complexity parameter $k_e$ (Vochezer et al., 2016). **(h)** and **(i)** show the particle shape fractions derived from the PPD-2K Fourier diffraction pattern analysis and machine learning diffraction pattern analysis, respectively. Arrows highlight the particle growth and sublimation periods with corresponding changes in the fraction of pristine ice crystals.





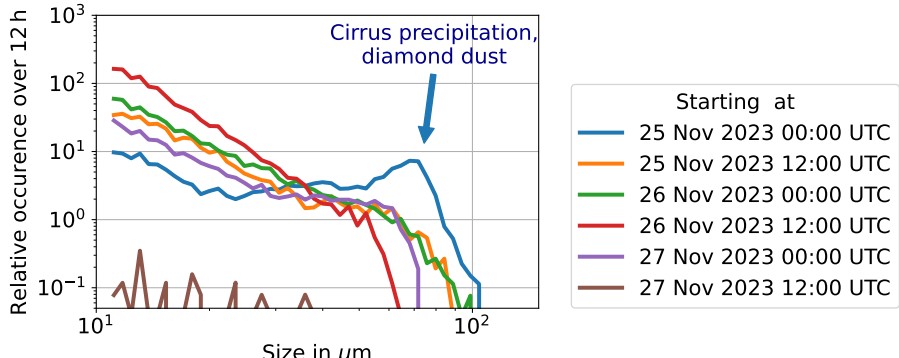

**Figure 6.** Temporal evolution of the particle size distributions integrated over 12 hours during the ice fog event of 25 November 2023. The particle mode with a modal maximum at particle sizes of 70 μm is observed only during the measurement period starting at 00:00 UTC on 25 November 2023, when streaks of ice particles were falling from high altitudes. For all 12-hour periods during the ice fog event the peak of the measured particle size distribution is at the lower size detection limit of PPD-2K at a spherical equivalent diameter of 11 μm. The last particle size distribution of 27 November 2023 12:00 UTC shows a low concentration at all bins, when no particles were seen on the lidar.

## 3.2 Ice fog event of 25 November 2023

From 25 to 28 November 2023 an event of boundary-layer ice particles was recorded with PPD-2K. The development and decay of this ground-level, thin cirrus cloud on the Antarctic plateau will be analysed in this section. Air parcel back-trajectories with the NOAA HYSPLIT model are shown in Fig. 4a. It can be seen in the 84 h back-trajectories at heights of 50 m, 200 m and 1000 m above ground that the air masses originate from inland Antarctica and do not have maritime influences. This confirms that the boundary layer ice crystals were locally formed on the Antarctic plateau through radiative cooling. Aerosol pollution can also be excluded due to the southerly winds during the event. Exhaust emissions from generators and vehicles operated at Concordia station were transported away from the measurement site. Details are provided in Appendix A.

Fig. 5 shows the temporal overview of the atmospheric conditions and the microphysical properties of the ice fog event observed between 25 and 28 November 2023. Prior to the event a period of high-level streaks of ice crystal precipitation originating from a height of more than 1000 m above ground ended on 25 November 2023 at around 15:00 UTC and a short period of clear sky started. This can be seen in the lidar backscattering signal in Fig. 5a that shows the vertical atmospheric profile at Dome C. At 18:00 UTC a thin layer of ice particles with a vertical extent of a few tens of meters started forming at ground level. The corresponding backscattering linear depolarisation ratio was larger than 0.1 indicating ice (see Fig. 5b). At the same time a pronounced temperature inversion of 10 K between the temperature of -34 °C at 42 m and a temperature of -44 °C at 3 m altitude above ground can be seen in Fig. 5d. At these low temperatures no supercooled liquid droplets could exist near ground. Therefore, all cloud and fog particles at ground level must have been frozen, which is confirmed by the low fraction of spherical particles seen in Fig. 5g and 5h.



From 18:00 UTC to 22:00 UTC the particle concentration at ground level measured with the PPD-2K gradually increased from $10^{-2}\,\mathrm{L}^{-1}$ to about $10^1\,\mathrm{L}^{-1}$ (see Fig. 5c). The confidence interval (one standard deviation) given in Fig. 5c combines the

systematic measurement uncertainty in concentration due to the uncertainty in the pump flow and the statistical error of the mean. Between about 15:00 UTC and 21:00 UTC the measured small-scale optical complexity $k_\mathrm{e}$ increased from values around 4.2 to 4.9 before decreasing back to around 4.3. This can be seen in the 300 s rolling mean of $k_\mathrm{e}$ in Fig. 5h. The confidence interval is the 300 s rolling standard deviation. To be statistically robust, $k_\mathrm{e}$ is only shown when the particle concentrations exceeds $10^{-1}\,\mathrm{L}^{-1}$, which is equivalent to five diffraction pattern images per 300 s mean.

Simultaneously to the increase in $k_\mathrm{e}$, the fraction of rough particles from the machine learning classification, characteristic for high surface roughness (Schmitt et al., 2024a), temporarily increased to about 50 % (see Fig. 5h). This sudden, transient increase in ice particle complexity is indicative of homogeneous freezing, as homogeneously frozen ice particles recorded with SID-3 have been associated with increased surface roughness (Ulanowski et al., 2014; Schnaiter et al., 2016). The statistical significance between the increased $k_\mathrm{e}$ values recorded between 15:00 UTC and 21:00 UTC on 25 November 2023 (315 particles

with mean $k_\mathrm{e}$ of 4.69) in comparison to the $k_\mathrm{e}$ values recorded in six-hour time periods before 15:00 UTC (1562 particles with mean $k_\mathrm{e}$ of 4.18) and after 21:00 UTC (9784 particles with mean $k_\mathrm{e}$ of 4.26) has been confirmed with two-sample Welch t-tests (p<0.01).

After 26 November 2023 00:00 UTC the ice particle layer showed a higher lidar backscattering signal than the previously recorded high-level precipitation confirming the higher particle concentration measured with PPD-2K (see Fig.5a). It grew

to a vertical extent of more than 200 m, likely caused by convective boundary layer growth, with a backscattering linear depolarisation higher than 0.1 (see Fig.5b). Simultaneously, we see the particle growth by an increase in particle size (see particle size distribution in Fig.5c). The particle growth correlates with an increase in the fraction of pristine ice particles, shown in Fig. 5g,h. Fig. 4b shows a photo of the PPD-2K measurement site at 00:19 UTC on 26 November 2023. The red arrow above the horizon highlights a thin fog layer. Due to the decreased visibility the event of low-level boundary layer ice

crystals can be classified as an ice fog event (American Meteorological Society, b; World Meteorological Organization, b). The lack of other possibly seeding clouds at higher altitudes on the lidar backscattering signal supports this classification. The maximum height was reached at about 07:00 UTC on 26 November 2023 and the vertical extent of the ice fog layer started to decrease again. The particles sublimated with a visible reduction in particle size (see Fig.5c), a reduction in the fraction of pristine ice particles and an increase in the fraction of sublimating particles (see Fig. 5g,h). At the same time the

lidar backscattering signal showed a decay until 10:00 UTC, when the signal started to increase again. The increase in lidar backscattering signal and particle concentration at around 10:00 UTC on 26 November 2023 coincides with a temperature inversion building up again (see Fig. 5a,c,d,e). Simultaneously, the particle size increases, the fraction of pristine ice particles increases and the fraction of sublimating particles decreases again. At 18:00 UTC on 26 November 2023 the ice fog layer started to weaken with a decrease in particle size and fraction of pristine ice particles. The ice fog layer disappeared in the

lidar backscattering signal for a few hours starting at around 00:00 UTC on 27 November 2023 (see Fig.5a,b). There was a simultaneous drop in the particle concentration measured with PPD-2K (see Fig. 5d). The ice fog layer then reappeared from 04:00 UTC to 11:00 UTC with a lower $\delta$ of about 0.1 (see Fig.5a,b). Then the final decay can be seen with a drop in particle





concentration of more than two orders of magnitude from more than $1\,\mathrm{L}^{-1}$ to about $10^{-2}\,\mathrm{L}^{-1}$. Between 00:00 UTC on 26 November 2023 and 00:00 UTC on 27 November 2023 the diurnal cycle weakly affected the development of the ice fog event.

It is interesting to note that the strong inversion period at about 18:00 UTC on 27 November 2023 does not initiate ice fog. A possible explanation is that the ice fog was nucleated at a different location in the vicinity with more favourable nucleation conditions and was subsequently transported to the measurement site. This can also explain the temporary reappearance of the ice fog around 06:00 UTC on 27 November 2023.

During the time frame of the ice fog event only small changes in wind direction occurred. During the ice fog event, wind

speeds were mainly between $6\,\mathrm{m/s}$ and $8\,\mathrm{m/s}$, slightly higher than the $4\,\mathrm{m/s}$ to $6\,\mathrm{m/s}$ observed before and after the event. The wind direction moved from about 320° before the ice fog event to about 250° during and after the event. For this range of wind directions only clean air is transported to the measurement site with no possible pollution from the research station (see Appendix A). For wind speeds exceeding $7\,\mathrm{m/s}$ a contribution of blown snow cannot be excluded. Nonetheless, between 00:00 UTC and 10:00 UTC on 27 November 2023 the particle concentration drops to about $5\cdot10^{-1}\,\mathrm{L}^{-1}$ during clear atmo-

spheric conditions at wind speeds of about $7\,\mathrm{m/s}$. This can be seen as an upper concentration limit for a possible contribution of blown snow at wind speeds of about $7\,\mathrm{m/s}$.

Fig. 6 shows the temporal evolution of particle size distributions measured with PPD-2K from 00:00 UTC on 25 November to 00:00 UTC on 28 November 2023. Each particle size distribution represents a 12-hour integration of Fig. 5c, starting at the labelled measurement time. The particle size distribution starting at 00:00 UTC on 25 November 2023 shows the particle

size distribution during an event of streaks of ice crystals falling from altitudes of more than $1000\,\mathrm{m}$ above ground, preceding the onset of the ice fog event (see Fig.5a and Fig. 5b). The streaks of falling ice particles are cirrus precipitation or, when occurring without overlying clouds, frozen clear sky precipitation (also known as diamond dust) (American Meteorological Society, a; World Meteorological Organization, a). The cirrus precipitation and diamond dust event exhibits a bi-modal particle size distribution with one peak at the lower size detection limit of PPD-2K at spherical equivalent diameters of $11\,\mu\mathrm{m}$ and a

second peak at spherical equivalent diameters of about $70\,\mu\mathrm{m}$.

For the measurement times when the ice fog is seen on the lidar backscattering signal, only the mode with the peak at the lower size detection limit of PPD-2K is observed in Fig.6b-e. This mode with the peak at the lower size detection limit of PPD-2K at $11\,\mu\mathrm{m}$ is hereafter referred to as ice fog mode. The second mode with the peak at spherical equivalent diameters of about $70\,\mu\mathrm{m}$ is only seen in Fig.6a, associated with fall streaks of ice particles from high altitudes. This mode is hereafter

called cirrus precipitation - diamond dust mode (cpdd mode). It is absent in Fig.6b-f. Because the cirrus cloud cover during the event of fall streaks is usually not continuous and ice particles are detected at times of clear sky too, the class of cirrus precipitation is combined with diamond dust.

The distinct difference in particle size distribution of the ice fog event and the cirrus precipitation - diamond dust (cpdd) events motivates the combined analysis of the microphysical properties of all ice fog and cpdd events that occurred during the

measurement campaign at Dome C.



### 3.3 Microphysical properties of different precipitation events

In this section the microphysical properties such as particle habit, concentration, size distribution and small-scale optical complexity ($k_{\mathrm{e}}$) of ice particles measured with PPD-2K are compared for ice fog events (labelled: if) and cpdd events at Dome C, Antarctica. The events are identified manually from the lidar backscattering signal (see Fig.5a). Ice fog periods exhibit
relatively constant low-level ice clouds, similar to Fig. 5a,b from 26 November 2023 00:00 UTC to 27 November 00:00 UTC. cpdd periods show streaks of ice crystals that are falling from high altitude with no liquid and no dense clouds present, similar to Fig. 5a,b between 00:00 UTC and 15:00 UTC on 25 November 2023. Out of the non-polluted operational periods 19.7 % are classified as cpdd events and 9.9 % are classified as ice fog events. The rest is prominently clear sky and occasional liquid cloud cover.

Two events were observed when ice fog and cirrus precipitation occurred simultaneously (2.0 % of the non-polluted operational period). These are not used for the classification. The complete measurement period from 21 November 2023 to 09 January 2024 is added (labelled: all). Measurement times are excluded when pollution from Concordia station was possible due to the wind direction. For comparison the microphysical properties of urban, polluted ice fog events at Fairbanks, Alaska (labelled: FB), recorded during three winters between 2020 and 2022, are added (Schmitt et al., 2024b). The ice fog data from
Fairbanks was also collected with PPD-2K.

Fig. 7a shows the particle size distribution of the different particle events. All events exhibit the ice fog mode, characterised by a pronounced increase towards the lower size detection limit of PPD-2K at a spherical equivalent diameter of 11 µm. The particle size distributions from all non-polluted data from Dome C and the particle size distributions of the cirrus precipitation and diamond dust events from Dome C additionally show the cpdd mode with a peak at a spherical equivalent diameter of
approximately 70 µm. This clearly distinguishes ice particles formed near the surface during ice fog events from particles falling in streaks from altitudes exceeding 1000 m. During the austral summer at Dome C, cpdd events dominate the boundary layer ice particles with sizes larger than about 50 µm and ice fog dominates for particle sizes smaller than about 40 µm .

Furthermore, it can be noted that the particle size distribution of the urban, polluted Fairbanks ice fog sharply drops for particle sizes of about 50 $\mu m$. This is not the case for the mean particle size distribution of the Dome C ice fog where the
particle concentration gradually drops for particle sizes between 50 $\mu m$ and 100 $\mu m$. The sharper drop for ice fog at Fairbanks can be attributed to the high aerosol concentration. The higher aerosol concentration prevents ice particles from growing to larger sizes as they do in the clean environment on the Antarctic plateau due to water vapour competition.

The statistical analysis of the measured particle concentration is shown in Fig. 7b. The mean particle concentration is the mean of all 600 s time averages that occurred during the lidar classified particle events. The mean particle concentration during
the ice fog events at Dome C is with $3.8\,\mathrm{L^{-1}}$ about four times higher than during the cpdd events at Dome C with $9.6 \cdot 10^{-1}\,\mathrm{L^{-1}}$. Ice fog is known for its higher particle concentration in comparison to diamond dust events (Girard and Blanchet, 2001a). The mean particle concentration of the complete measurement period excluding pollution times is with $3.1\,\mathrm{L^{-1}}$ higher than during the cpdd events and lower than during the ice fog events. The ice fog at Fairbanks, Alaska has the highest mean particle concentration with $3.5 \cdot 10^{2}\,\mathrm{L^{-1}}$. This is caused by anthropogenic water vapour emissions and high aerosol load at this urban





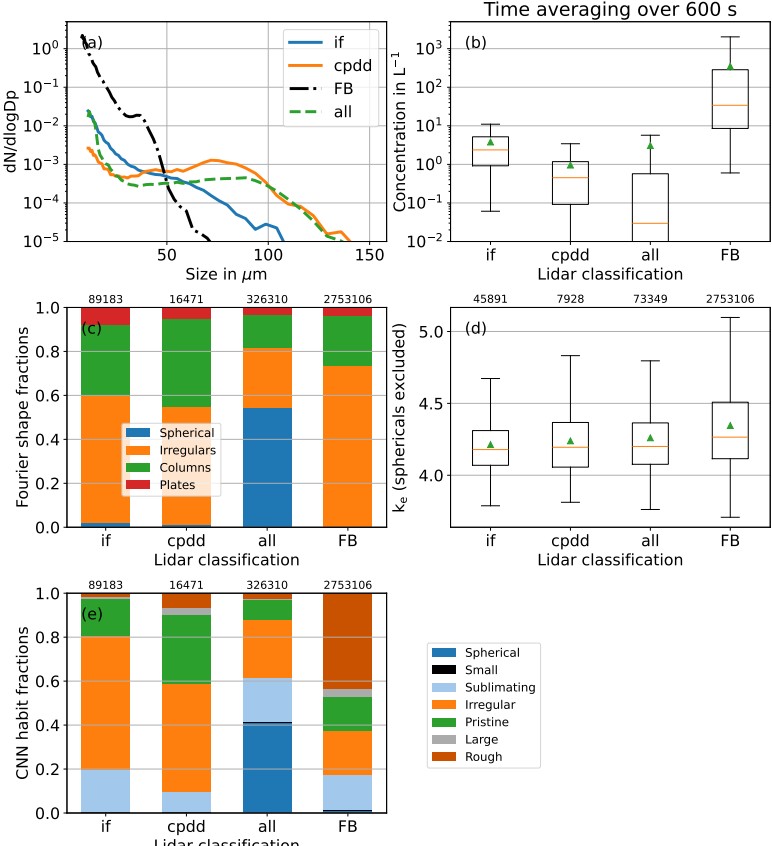

**Figure 7.** Ice particle microphysical properties for periods with low-level ice clouds (labelled if for ice fog), for periods with high-level streaks of ice crystals (labelled cpdd for cirrus precipitation - diamond dust) and for the complete measurement period between 21 November 2023 and 09 January 2024 (labelled: all). Ice fog and cpdd events are identified from a manual classification of the lidar data. Urban, polluted ice fog events from a three-year study between 2020 and 2022 at Fairbanks, Alaska is shown for comparison (labelled: FB) (Schmitt et al., 2024b). **(a)** shows the mean particle size distributions (dN/dlogDp), **(b)** shows the mean particle concentration of particles larger than 11 µm, **(c)** shows the habit fractions derived with the Fourier analysis by Vochezer et al. (2016), **(d)** shows the particle small-scale optical complexity ($k_e$) derived from the diffraction pattern speckle analysis (sphericals excluded) and **(e)** shows the habit fractions derived with the machine learning method by Schmitt et al. (2024a). Smaller and more complex particles occur with a higher concentration for the urban Fairbanks ice fog in comparison to the Antarctic ice fog. At Dome C, the cpdd events have an additional size mode at approximately 70 µm that is absent for the ice fog events.





| Averaging period | Maximum measured particle concentration | | | |
|---|---|---|---|---|
| | if | cpdd | all | FB |
| 600 s | $4.9 \cdot 10^1 \, \mathrm{L}^{-1}$ | $1.2 \cdot 10^1 \, \mathrm{L}^{-1}$ | $6.4 \cdot 10^2 \, \mathrm{L}^{-1}$ | $4.3 \cdot 10^3 \, \mathrm{L}^{-1}$ |
| 60 s | $5.6 \cdot 10^1 \, \mathrm{L}^{-1}$ | $3.6 \cdot 10^1 \, \mathrm{L}^{-1}$ | $1.1 \cdot 10^3 \, \mathrm{L}^{-1}$ | $2.1 \cdot 10^4 \, \mathrm{L}^{-1}$ |

**Table 2.** Maximum measured particle concentration of particles larger than 11 μm for time averaging of 60 s and 600 s during the ice fog (if) events, cirrus precipitation and diamond dust events (cpdd) and the complete measurement period (all) at Dome C. The data of the urban, polluted ice fog events from a three-year study between 2020 and 2022 at Fairbanks, Alaska are shown for comparison (FB). The higher peak concentrations at Fairbanks can be attributed to the high aerosol pollution and water vapour emissions at the urban measurement site.

location (Schmitt et al., 2013, 2024b). The high aerosol load creates a higher concentration of ice nucleating particles and the excess water vapour increases the relative humidity promoting ice crystal growth.

Table 2 shows the maximum particle concentration for time averages of 60 s and 600 s. The maximum concentration during the complete Antarctic data set is $1.1 \cdot 10^3 \, \mathrm{L}^{-1}$ over 60 s. It is higher than the maximum particle concentrations during the lidar classified cpdd and ice fog event with maximum concentrations of $3.6 \cdot 10^1 \, \mathrm{L}^{-1}$ and $5.6 \cdot 10^1 \, \mathrm{L}^{-1}$ over 60 s. This can be

attributed to two high-concentration, low-level supercooled liquid clouds events that occurred on 17 December 2023 and 18 December 2023. These events of liquid fog will be analysed in detail in a future work. The ice fog from Fairbanks has peak concentrations about one order of magnitude higher than the Antarctic ice fog measurements for both averaging periods. This is expected due to the high air pollution and anthropogenic water vapour emission at the measurement site in Fairbanks. It needs to be taken into account that PPD-2K only measured the concentration of particles larger than the lower detection size

limit of 11 μm.

Fig. 7c shows the particle fractions with plate-like, columnar, irregular and spherical diffraction patterns identified with the Fourier analysis of Vochezer et al. (2016). The fractions are similar with about 58 % and 54 % irregulars, 32 % and 40 % columns, 8 % and 5 % plates and 2 % and 1 % sphericals for all ice fog events and cpdd events at Dome C, respectively. This is an unexpected finding because the ice fog and cpdd particles have different growth regions. Of the complete measurement

period 28 % had irregular, 15 % had columnar, 3 % had plate-like and 54 % had spherical diffraction patterns. The high fraction of spherical particles in is predominantly caused by two high-concentration, low-level supercooled liquid clouds events that occurred on 17 December 2023 and 18 December 2023, which will be analysed in separate publication. The ice fog events at Fairbanks, Alaska have 73 % irregulars, 23 % columns, 4 % plates, and 0 % sphericals. The higher fraction of irregular particles is likely caused by the high industrial water vapour emission at Fairbanks that promote faster ice crystals growth. More water

vapour increases particle growth speed and complexity (Bailey and Hallett, 2009; Schnaiter et al., 2016).

The small-scale optical particle complexity is characterised with the $k_e$ parameter derived from the PPD-2K diffraction patterns applying the speckle texture analysis (see Fig. 7d). The mean small-scale complexity of the Dome C ice fog events, cpdd events and the complete measurement period at Dome C is similar with means of 4.21, 4.24 and 4.26, respectively. The overall low particle complexity independent of the precipitation event can be attributed to the cold, clean and dry environment

on the Antarctic plateau where slow growth conditions persist promoting the growth of pristine ice particles. The ice fog





particles measured at Fairbanks, Alaska show a higher mean complexity $k_e$ of $4.35$. This higher particle complexity can also be attributed to more water vapour and faster crystal growth in the atmosphere at Fairbanks.

In Fig. 7e the particle fraction of the machine learning classification by Schmitt et al. (2024a) is shown. It gives additional insights due to the different habit classes in comparison to the Fourier analysis method. 19 % of the ice fog particles from
Dome C and 16 % of the ice fog particles from Fairbanks, Alaska have sublimating diffraction patterns, while the cpdd events at Dome C have a lower fraction of sublimating particle of 10 %. The fraction of pristine particles during the cpdd events at Dome C is with 31 % higher than during the Dome C ice fog events with 17 % and the Fairbanks ice fog with 15 %. This means that a larger fraction of particles during the cirrus precipitation and diamond dust events has a prominent maximum at the 22° halo scattering angle in comparison to the ice particles during the ice fog events. A possible reason can be different growth
conditions for the cirrus precipitation and diamond dust particles that fall in streaks from higher altitudes in comparison to the ice fog particles that grow near ground level. Furthermore, the ice fog at Fairbanks, Alaska has a high fraction of 43 % rough particles in comparison to 2 % for the Antarctic ice fog events. This supports the hypothesis of more complex ice particles in Fairbanks due to more available water vapour during particle growth from anthropogenic water vapour emission.

## 4 Discussion

We compared the ice crystal morphology of ice fog to cirrus precipitation and diamond dust events. Since existing definitions do not clearly separate ice fog and diamond dust, we manually identified ice fog events using lidar-based identification, applying the definitions of the American Meteorological Society and the World Meteorological Organisation. Both define ice fog as a fog composed of suspended small ice particles and diamond dust as ice crystals falling from a cloudless sky (American Meteorological Society, b; World Meteorological Organization, b). The distinct differences that we see in particle morphology
of ice fog and cpdd events confirm the lidar-based separation of the boundary layer ice particle phenomena.

The cpdd mode with a modal maximum of approximately $70\,\mu m$ is in good agreement with the diamond dust size parametrisation by Girard and Blanchet (2001a) that states a lower size limit of $30\,\mu m$. Our results of cirrus precipitation and diamond dust particle size distributions are also comparable to findings for diamond dust by Lawson et al. (2006) at South Pole station with the Cloud Particle Imager (CPI). They used the term diamond dust for precipitation, while scattered, thin clouds were
present, similar to our cpdd class. It needs to be taken into account that the particle maximum dimension can be 1 to 2.5 the spherical equivalent diameter measured with PPD-2K. The cpdd events exhibit a mean particle concentration of $9.6\cdot10^{-1}\,L^{-1}$, which is within the expected range of concentrations lower than $4\cdot10^{3}\,L^{-1}$ by Girard and Blanchet (2001a). Yet, it is lower than diamond dust concentrations measured at South Pole Station with CPI of up to $10^{3}\,L^{-1}$ by Lawson et al. (2006). Overall, the microphysical properties of the cpdd events align with previous measurements of diamond dust. Therefore, the effect of
the cirrus precipitation or seeding on the diamond dust microphysical properties in the particle size range of PPD-2K is likely limited.

The ice fog mode falls within the range of expected particle diameters smaller than $30\,\mu m$ by Girard and Blanchet (2001a). During ice fog events we observed a mean concentration of $3.8\,L^{-1}$. This is lower than the concentration expected by Girard



and Blanchet (2001a) of at least $1 \cdot 10^3 \, \text{L}^{-1}$. The lower-than-expected particle concentration of the Antarctic ice fog events can
have multiple reasons.

Firstly, the modal maximum seems to be below the lower size detection limit of PPD-2K due to the strong increase in particle concentration for the smallest observable sizes. Therefore, we are missing all particles with sizes smaller than 11 μm that can have a significant contribution to the particle concentration. The real particle concentration, including the smallest ice particles, is likely higher than our PPD-2K measurements. This may also explain why Schmitt et al. (2024b) found lower particle
concentrations with PPD-2K than in previous measurements with Formvar replica and video-imaged particles (Schmitt et al., 2013). Secondly, the Antarctic plateau can be considered the cleanest environment on the planet (Heumann, 1993). There is a very low aerosol concentration on the ice cap at high altitudes inside the polar vortex (Virkkula et al., 2022). Aerosol and water vapour emissions from anthropogenic activity and open water are almost non-existent. Thus, low ice particle concentrations are expected (Maciel et al., 2023). The dry and clean environment significantly differs from the Arctic, where intrusions of
non-polar air masses and open water are much more common (Waugh and Randel, 1999). Therefore, we suggest that in situ ice fog on the Antarctic plateau can have a lower particle concentration than $1 \cdot 10^3 \, \text{L}^{-1}$. To define the exact concentration measurements of particle sizes down to a few microns are needed.

The nucleation process of in situ ice fog on the Antarctic plateau remains a topic of current research. With the measured ice particle morphology for ice particles with sizes down to 11 μm at Dome C we can contribute further information to the open
question. Ice particles can nucleate via two pathways. The first pathway is homogeneous nucleation, where supercooled liquid droplets freeze spontaneously at temperatures below approximately -38 °C (Heymsfield et al., 2017). The exact temperature depends on the droplet size and water activity (Koop et al., 2000). The second pathway is heterogeneous freezing, in which the ice particles nucleate assisted by an ice nucleating particle. Multiple mechanisms of heterogeneous freezing exist (Heymsfield et al., 2017). According to Gultepe et al. (2017) both pathways are possible for the nucleation of ice fog. In inland Antarctica
a low concentration of ice nucleating particles persists due to continuous snow cover, very limited biological activity and atmospheric isolation due to the strong Antarctic polar vortex (Belosi et al., 2014; Sauerland et al., 2024). Vignon et al. (2022) concluded from lidar, temperature and humidity measurements during an ice fog event at Dome C that the ice fog was homogeneously nucleated, but no direct evidence was found. Our observations provide further support for the hypothesis of homogeneous freezing. The first indication is the initiation of the ice fog event of 25 November 2023 with a short-term
significant increase in small-scale optical complexity reaching $k_e$ = 4.9. This findings support homogeneous freezing as a possible nucleation pathway for locally formed ice fog events on the Antarctic plateau, as suggested by Vignon et al. (2022). For further analysis of the nucleation process simultaneous measurements of the cloud particles and the concentration of ice nucleating particles are needed. Comparing the concentration of ice nucleating particles before the initiation of the ice fog event to ice particle concentration during the ice fog event can provide further information.

The ice fog microphysical properties at Dome C differ from the urban, polluted ice fog in Fairbanks, Alaska. The mean particle concentration in Dome C ($3.5 \cdot 10^2 \, \text{L}^{-1}$) is more than an order of magnitude lower in comparison to Fairbanks ($3.8 \, \text{L}^{-1}$). This can be attributed to more available water vapour in Fairbanks from burning fossil fuels for heat and power generation in the urban region. The simultaneously higher aerosol load, and thus higher concentration of ice nucleation particles, allows



more particles to nucleate heterogeneously but limits their maximum sizes due to water vapour competition. The larger amount of water vapour furthermore favours a higher growth speed, which causes more complex ice crystals to occur at Fairbanks in comparison to Dome C. This results in a higher mean $k_e$ value of $4.35$ in comparison to $4.26$ at Dome C and a with $43\,\%$ much larger fraction of rough particles in comparison to $2\,\%$ at Dome C according to the machine learning classification. Similarly rough ice particles at high concentration were measured during test operation of PPD-2K next to Concordia station's generator exhaust plumes (not shown here).

The measurements of ice crystal size, concentration, shape, and complexity of boundary layer ice particles on the Antarctic plateau are important for the validation of satellite remote sensing due to the distinct differences from the Arctic. The next step would be to deploy an instrument year-round to identify the boundary layer ice particle morphology during winter time. This can only be done with an instrument adapted to withstand the extremely cold winter conditions at Dome C, where temperatures can drop to -80 °C. In this work the daily sun hours at Fairbanks, Alaska were below a maximum of 7 h per day while the measurements at Dome C occurred during polar day. Year-round measurements can enable a comparison between Fairbanks, Alaska and Dome C at similar solar radiation. Furthermore, measurements of particle concentration for size distributions down to single micrometers would be useful to identify the modal maximum of the ice fog particle size distribution and to properly measure the ice fog particle concentration over the complete size mode. Additional measurements of the scattering phase function could give further insight on the radiative forcing of low-level boundary layer ice particles in polar regions.

## 5  Conclusions

During the in situ measurements of ice crystal size and concentration at Dome C, we observed a particle mode at approximately $70\,\mu\text{m}$ for cirrus precipitation and diamond dust events, which was absent during ice fog events. For ice fog events a smaller particle mode dominated with a modal maximum below the PPD-2K size detection limit of $11\,\mu\text{m}$. The mean particle concentration during cpdd events ($9.6 \cdot 10^{-1}\,\text{L}^{-1}$) was lower than the mean of ice fog events ($3.8\,\text{L}^{-1}$). The observed particle sizes are consistent with the size parametrisation of Girard and Blanchet (2001a) for Arctic regions. The mean particle concentration of the investigated Antarctic ice fog was lower than the concentration parametrisation by Girard and Blanchet (2001a) and lower than the mean particle concentration during urban ice fog events in Fairbanks, Alaska ($3.5 \cdot 10^2\,\text{L}^{-1}$). Additionally, ice fog particles at Dome C were more pristine than at Fairbanks. This emphasises the difference between urban, polluted ice fog and ice fog in the clean environment of the Antarctic plateau and highlights the need for a separate concentration parametrisation of inland Antarctic ice fog. The microphysical properties measured at Dome C, Antarctica during austral summer are valuable for satellite remote sensing validation on the Antarctic plateau. In addition, they can be used to improve the microphysical parametrisation of boundary layer ice crystals in climate models of Antarctica, contributing to the few in situ measurements existing in this remote region.



*Data availability.* PPD-2K measurement data from Dome C, the depolarisation lidar data and the measurement tower temperature and
humidity data are available on https://doi.org/10.5281/zenodo.16419695 (version v2). The automated weather station data are available on
https://www.climantartide.it/.

**Appendix A:  Additional information of the ice fog event of 25 November 2023 and the fog event of 17 December 2023**

PPD-2K measurements are excluded for the times when the wind direction did not rule out pollution from the generators at
Concordia station. In Fig. A1a and Fig. A1b the wind direction and wind speed from the automated weather station at Dome
C are shown for the ice fog event of 25 November 2023. The areas of wind speed and wind direction are highlighted in blue
when aerosol pollution from the station's generators and motorized vehicles is possible (Virkkula et al., 2022). It can be seen
that no aerosol pollution is possible during the ice fog event.

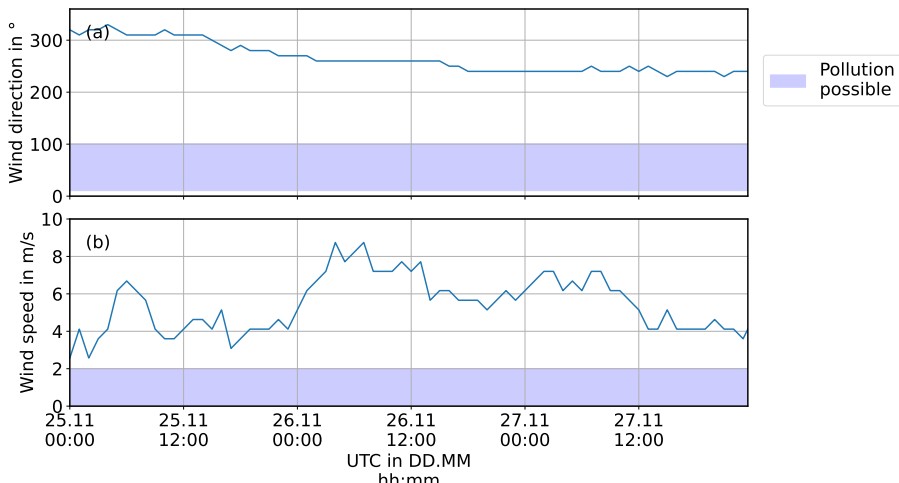

**Figure A1.** Wind direction **(a)** and wind speed **(b)** from the automated weather station at Concordia station for the ice fog event of 25
November 2023 (Grigioni et al., 2022). The measurements are not affected by the station's exhaust emissions. The areas highlighted in blue
mark wind speed and wind direction when aerosol pollution from the station's generators and motorized vehicles can occur (Virkkula et al.,
2022).

**Appendix B:  PPD-2K threshold of optical small-scale complexity $k_\mathrm{e}$ for rough particles**

Fig. B1 shows example diffraction patterns of columnar particles with the maximum Fourier coefficient of second order. They
are sorted by the small-scale optical complexity parameter $k_\mathrm{e}$. Similar to the SID-3 diffraction patterns in Schnaiter et al.
(2016), PPD-2K diffraction patterns show speckles outside of the columnar arcs for particles with $k_\mathrm{e}$ higher than about 4.6.
These particles are considered complex.



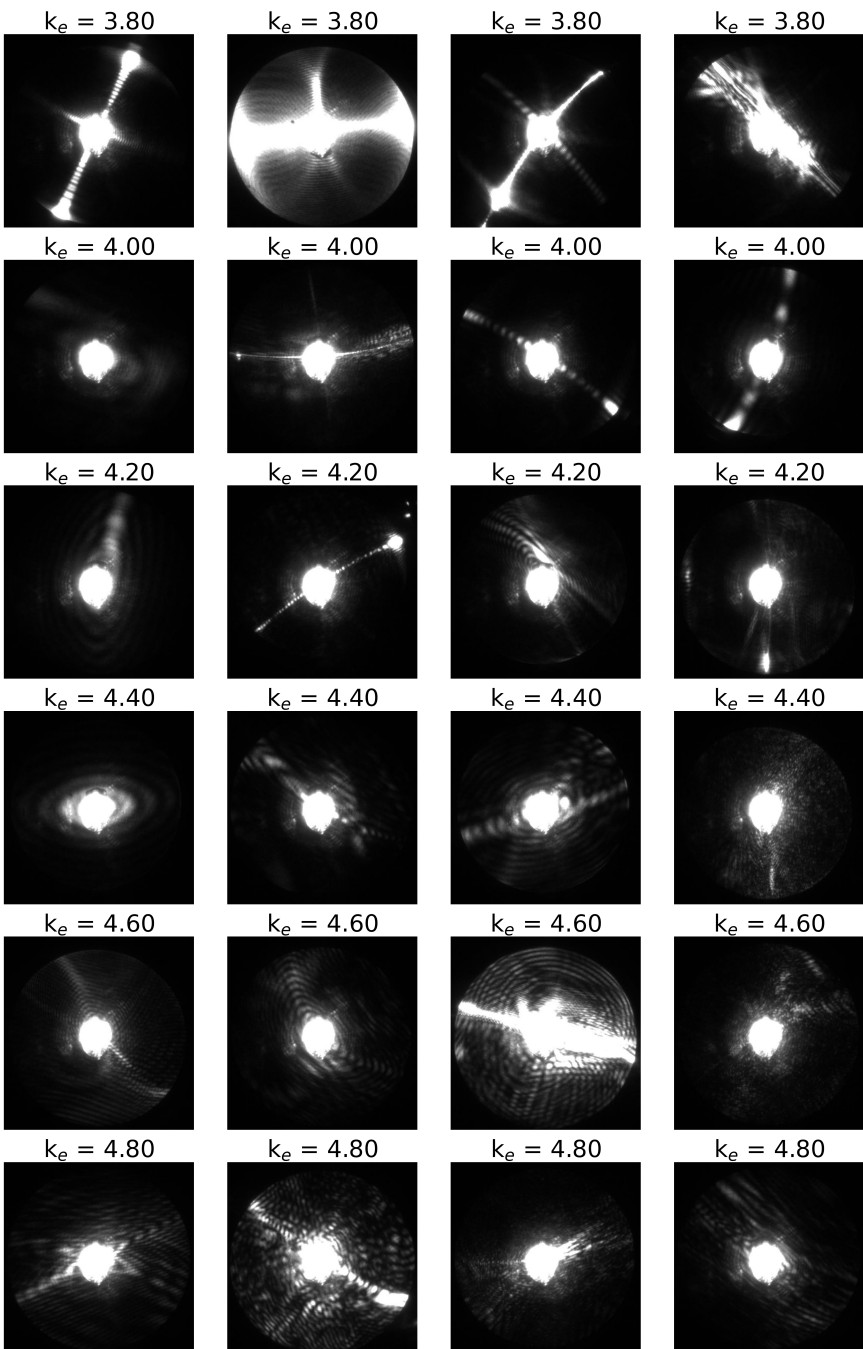

**Figure B1.** Example diffraction patterns of columns with the maximum Fourier coefficient of second order sorted by the small-scale optical complexity parameter $k_e$. Particles with $k_e$ of about 4.6 and higher show speckles outside of the columnar arcs and are considered complex.



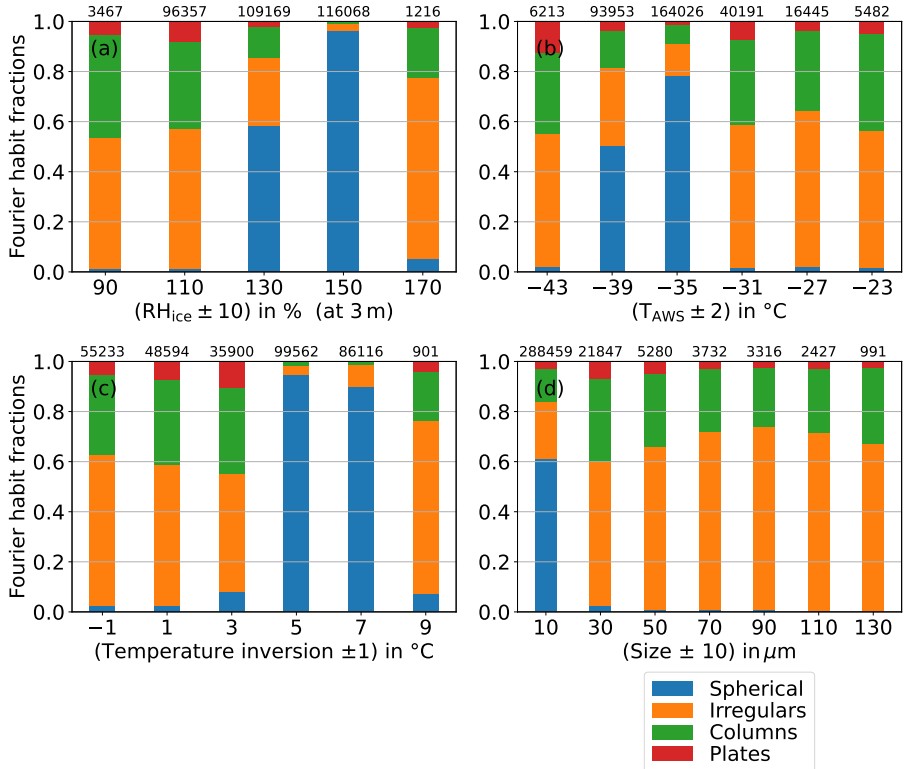

**Figure C1.** Particle fractions determined with the Fourier method (Vochezer et al., 2016) for different ranges of relative humidity with respect to ice ($RH_{\mathrm{ice}}$) **(a)**, automated weather station temperature ($T_{\mathrm{AWS}}$) **(b)**, temperature inversion **(c)** and particle size **(d)**. The temperature inversion is defined as the difference in air temperature measured at 42 m above ground and at 3 m above ground at the meteorological tower of Concordia station. Only non-pollution times are included in the analysis.

## Appendix C:  Habit fractions for different atmospheric conditions

Fig. C1 shows the habit fractions determined with the Fourier analysis of the diffraction patterns (Vochezer et al., 2016).
Comparing the Fourier and the machine learning methods similar trends for pristine and spherical particles can be noted. But there are also differences. For example, the Fourier method detects a slightly higher fraction of spherical particles. It also detects a higher fraction of pristine particles for particle sizes larger than 40 μm in comparison to the combined fraction of columns and plates classified with the Fourier analysis.



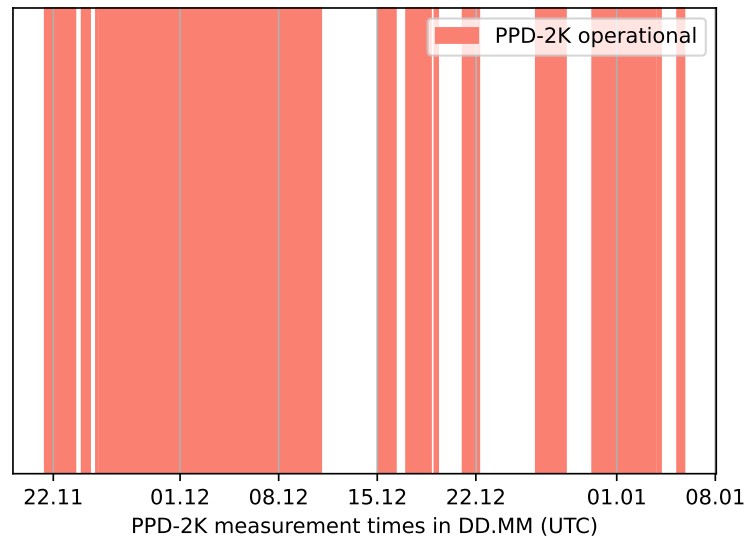

**Figure D1.** In red: Operational times of the PPD-2K on the roof of the physics shelter during the deployment at Dome C, Antarctica between November 2023 and January 2024. During the white periods PPD-2K was not operational due to servicing and instrument connection errors.

## Appendix D:  Operation time of PPD-2K at Dome C, Antarctica

Fig. D1 shows in red the operational times of PPD-2K during the deployment at Dome C, Antarctica between November 2023 and January 2024. During the white periods PPD-2K was not operational due to servicing, maintenance and instrument connection errors.



*Author contributions.* AH and MS conceptualised the manuscript. MDG operated the lidar and webcam at Dome C. CS operated PPD-2K during the measurements in Fairbanks, Alaska and contributed the machine learning habit classification. CG operated the temperature and
humidity measurements at the meteorological tower at Dome C. AH, MS, EJ, CS and MDG interpreted the data. AH operated the PPD-2K at Dome C, analysed the data and wrote the manuscript with contribution of MDG, CS, CG, EJ and MS.

*Competing interests.* None of the authors declare competing interests.

*Acknowledgements.* This work was funded by the Helmholtz Association's Initiative and Networking Fund, grant number VH-NG-1531 and the Italian Antarctic Program (PNRA), project PNRA18-00058 'ICE-OPT'. The authors gratefully acknowledge the NOAA Air Resources
Laboratory (ARL) for the provision of the HYSPLIT transport and dispersion model and READY website (https://www.ready.noaa.gov) used in this publication. Concordia station is jointly operated by the French IPEV and Italian PNRA polar institutes. The CALVA project supported by IPEV (program #1013) provided meteorological including water vapour supersaturation observation records. A research travel to Fairbanks, Alaska was funded by the GRACE graduate school of the KIT Climate and Environment Center. The authors are grateful to the Concordia station staff for their support during the measurements.



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
