# Peer review of "Unique Microphysical Properties of Small Boundary Layer Ice Particles under Pristine Conditions on Dome C, Antarctica"

_EGUsphere, 2025_

## Referee Comment (RC1)

**Review of 'Unique Microphysical Properties of Small Boundary Layer Ice Particles under Pristine Conditions on Dome C, Antarctica'**

by Adrian Hamel et al.

This article leverages original measurement data from a PPD-2K instrument at Dome C that enables a characterization of ice crystal optical and geometrical properties falling above the Antarctic Plateau. A statistical analysis of ice crystal size and habits over a 2 month period is presented and a particular event of ice fog and diamond dust is analysed in details with the use of additional measurements collected at the meteorological observatory. A comparison with similar measurements collected in Alaska is also presented and discussed, and the paper concludes on the uniqueness of the ice crystal properties in the pristine environment of the Antarctic Plateau.

This is a quite interesting and original study with a serious data processing and analysis. The PPD-2K offers very precious and new information on the microphysical properties of Antarctic fog. The paper is overall well written and has the potential to become a relevant contibution to the scientific literature but I think some work is needed to make the analysis and conclusions more robust before its publication in Atmospheric Chemistry and Physics.

**Major** **comments :**

- The measurements cover a summer period, where the Dome C atmospheric boundary layer exhibits a marked diurnal cycle (e.g., Genthon et al. 2010, https://doi.org/10.1029/2009JD012741
). An analysis of the possible influence of local time on the occurrence of fog and diamond dust events would be very valuable.

- Albeit infrequent, blowing snow can occur at Dome C (drifting snow is however quite common).
A more in depth analysis of the possible 'contamination' of the crystal properties's analyses by blowing snow is absolutely needed. Scatter plots showing the crystal concentration as a function of wind speed, Particle size distributions and CNN habit fractions for different wind speed classes, and wind speed time series during the analysed fog events are examples of graph that could be helpful to address this point. In particular, one may wonder to what extent the fog cloud event analysed in Fig 5 and

Sect. 3.2 is not the remobilization of ice particles (from the diamond dust event a few hours before) by the wind.

- It is stated in the paper (l259) that the diurnal cycle boundary layer dynamics only weakly affects the development of the fog. I may be wrong but Fig 5 suggests the contrary to me. The fog layer deepens in local morning associated with the growth of the boundary layer (convective activity, as seen by the vertically homogeneous temperature in panel e) and the optical thinning near local noon and sharp increase in the relative fraction of sublimating particles might suggest a fog sublimation through vertical downward transport of dry air from the top of the convective boundary layer.

- At several places in the paper (especially in the discussion), the authors mention the homogeneous freezing process. However the distinction between homogeneous freezing of – relatively large – pre-existing supercooled liquid water (SLW) droplets at ~ -38°C and the homogeneous nucleation of ice through freezing of aerosol solution particles (at temperature below -38°C) should be distinguished. Vignon et al. 2022 suggested that the ice fog they observed was formes by homogeneous nucleation (freezing of small aerosol droplets), as the events they focused on took place in winter during which the temperature is too cold for SLW to exist at Dome C. Please distinguish the two processes during the analysis and clarify the text because it is confusing at some places and the interpretation might be not always correct.

- This is a comment related to the previous one. I find really unfortunate that the authors do not show an analysis of a liquid fog event (which could possibly lead to homogeneous freezing of supercooled liquid water drops). They mention that this is left for a future work, but the current paper is not that long and can include an additional case study. Such an additionnal analysis would allow to strengthen the interpretation of the spherical habits in relation with relative humidity wrt liquid, and to reinforce the conclusions about the occurrence of homogeneous freezing (of liquid droplets) at Dome C and its role in ice fog formation in summer. I would even say that the paper cannot keep the current message regarding the occurrence of homogeneous freezing without an additional investigation of liquid fog. Please see Ricaud et al. 2025 (https://doi.org/10.1016/j.polar.2025.101256) for additional information of nocturnal liquid fog at Dome C.

- I overall find the interpretation of the small-scale optical complexity parameter k not easy to follow at many places in the paper. Can you provides details about the physical meaning of this parameter and help the reader interpret its evolution.

**Minor comments :**

l12-13 : This is a quite strong statement given the conclusions from this study only holds for a limited period of time at a single location in Antarctica.

l16 'a fraction of about 40 % clear-sky precipitation' : this number is one estimate from one particular location. The contribution of clear-sky precipitation to the overall Antarctic precipitation is still an open question.

L33 : 'in a commonly used parameterisation' : are you sure the Girard & Blanchet parameterisation is 'commonly used' ? To my knowledge, I am not aware of many studies leveraging their microphysical parameterisation.

L37 : 2.85 and 2.57 K ? Where do those numbers come from ? How can such close numbers come from very different radiative forcings ? Girard & Blanchet (2001b) quantify the reduction in cooling rate associated with diamond dust and fog. The two types of cloud can lead to very different surface warming only if integrating over different time lengths.

Figure 1 : 'Time of flight in a.u.' can you specify the meaning of this label in figure's caption ?

L112 '10%' How has this number been estimated ?

L130-132 : What should the reader conclude about the difference of maintenance between the two periods ? Should we expect differences in data quality ?

L144 : The proper reference for this 7 m/s wind speed threshold to detect the occurrence of drifting snow is Libois et al. 2014
Libois, Q., Picard, G., Arnaud, L., Morin, S., and Brun, E.: Modeling the impact of snow drift on the decameter-scale variability of snow properties on the Antarctic Plateau, J. Geophys. Res.-
Atmos., 119, 662–681, https://doi.org/10.1002/2014JD022361,2014.

L148 : around → round

L149 aerosol → aerosols

L163 : 'regularly occurs' Please refer to Genthon et al. 2017 .

Genthon, C., Piard, L., Vignon, E., Madeleine, J.-B., Casado, M., and Gallée, H.: Atmospheric moisture supersaturation in the near-surface atmosphere at Dome C, Antarctic Plateau, Atmos.
Chem. Phys., 17, 691–704, https://doi.org/10.5194/acp-17-691-2017, 2017.

L163 : for the calculation of saturation vapor pressure.

L181 : The last sentence is awkward. Please reformulate.

L185 and Figure 3c : an additional panel with the relative humidity wrt liquid in x-axis would be helpful.

Figure 5f : please add the time series of RHliq as well. This might give insights into the ice nucleation mechanism at play.

Sect. 3.2 Can you explain why you chose this particular event and not another. Additional motivation and justification are needed.

L210 : 'radiative cooling' : not necessarily. Vignon et al. 2022 show that ice fog formation can be triggered through local cooling of the air associated with turbulent mixing.

L254 : ' layer started to weaken' → ice fog layer depth started to decrease ?

L291 : Can you really qualify a fog event as a 'precipitation event' ?

L294 : please recall the considered time period here.

L302 : 'measurement times are excluded when pollution …' this is a repetition from the Methods section.

L344 : emission → emissions

Appendix : There are too many appendices for a quite short paper (4). In particular, Appendix A and D seem not absolutely critical and can be summarized in a few words in the main text.

---

## Author Comment (AC1)

**Authors' Responses to Anonymous Referee #1**

Review of 'Unique Microphysical Properties of Small Boundary Layer Ice Particles under Pristine Conditions on Dome C, Antarctica' by Adrian Hamel et al. This article leverages original measurement data from a PPD-2K instrument at Dome C that enables a characterization of ice crystal optical and geometrical properties falling above the Antarctic Plateau. A statistical analysis of ice crystal size and habits over a 2 month period is presented and a particular event of ice fog and diamond dust is analysed in details with the use of additional measurements collected at the meteorological observatory. A comparison with similar measurements collected in Alaska is also presented and discussed, and the paper concludes on the uniqueness of the ice crystal properties in the pristine environment of the Antarctic Plateau. This is a quite interesting and original study with a serious data processing and analysis. The PPD-2K offers very precious and new information on the microphysical properties of Antarctic fog. The paper is overall well written and has the potential to become a relevant contibution to the scientific literature but I think some work is needed to make the analysis and conclusions more robust before its publication in Atmospheric Chemistry and Physics.

*Response: We thank Anonymous Referee #1 for the positive evaluation and the feedback for improvement of the manuscript. Below, we provide responses to the individual comments and detail the corresponding modifications made in the revised manuscript.*

Major comments :

- The measurements cover a summer period, where the Dome C atmospheric boundary layer exhibits a marked diurnal cycle (e.g., Genthon et al. 2010, https://doi.org/10.1029/2009JD012741 ). An analysis of the possible influence of local time on the occurrence of fog and diamond dust events would be very valuable.

*Response: The suggested analysis was performed and the data was analysed separately for different times of the day: 6 hours of night, morning, day and evening.*

- *The 'night' time frame covers the periods from 3 hours before local solar midnight to three hours after local solar midnight.*

- *The 'day' time frame covers the period from 3 hours before local solar noon to three hours after local solar midnight.*

- *The time frame 'evening' covers the periods from 3 hours after solar midnight to three hours before solar noon.*

- *The time frame 'morning' covers the periods from three hours after solar noon to three hours before local solar midnight.*

*It can be seen in the figure below that there are only minor influences of the time of the day on the occurrence of ice fog or cpdd (cirrus precipitation and diamond dust) events. We have added the event of supercooled liquid fogs which strongly depends on the time of the day as a comparison. The fraction 'all' is shown to verify that the instrument downtimes do not bias the distribution of the measurement periods over the time of the day. This is confirmed by the equal distribution of measurement data during the time of the day. **(a)** shows all data including the periods with possible pollution from Concordia station and **(b)** shows only the non-polluted data. It can be noted that in 'all' of **(b)** that the pollution events are more common in the morning time periods of the day.*

[Figure]

*The figure is added as supplementary material to the manuscript.*

- Albeit infrequent, blowing snow can occur at Dome C (drifting snow is however quite common). A more in depth analysis of the possible 'contamination' of the crystal properties's analyses by blowing snow is absolutely needed. Scatter plots showing the crystal concentration as a function of wind speed, Particle size distributions and CNN habit fractions for different wind speed classes, and wind speed time series during the analysed fog events are examples of graph that could be helpful to address this point. In particular, one may wonder to what extent the fog cloud event analysed in Fig 5 andSect. 3.2 is not the remobilization of ice particles (from the diamond dust event a few hours before) by the wind.

*Response: We agree that a more in-depth analysis of the possible effect of blowing snow on the ice fog events is important. The figure below shows the measured particle concentration in (a) and (b), the particle size distribution in (c) and the habit classes (d) during the observed non-polluted ice fog events as a function of the wind speed. It shows that the in situ measurements of the ice particle microphysics near ground level depend on the wind speed, but no simple relation between wind speed and particle concentration or particle size distribution is observed (e.g. a monotone increase of particle concentration). This is indicated for example by the decrease of the mean particle concentration with wind speed for wind speeds between 6 m/s and 8 m/s as well as by the particle size distribution, which is rather independent of the wind speed in the range between 4 m/s and 8 m/s. However, the fraction of sublimating particles increases with increasing wind speed and the fraction of pristine particle decreases with increasing wind speed which can be an influence of blowing snow, as blowing snow ice particles are expected to have rounded shapes (Pomeroy, 1989). Furthermore, most ice fog particles at wind speeds between 4 m/s and 8 m/s and few ice fog particles were measured at wind speeds below 4 m/s. Therefore, a possible contribution of blowing snow to the ice fog events cannot be excluded. However, we suggest that a larger dataset over a longer time period is needed to quantify the contribution of blowing snow to summer time ice fog at Dome C.*

[Figure]

*The possible effect of blowing snow on the ice fog events is now mentioned in the manuscript in lines 143f.: "For high wind speeds blowing snow can occur on the Antarctic plateau. To have an increased particle count rate at the measurement height of PPD-2K the wind speed must exceed about 7 m/s as seen in the multi-year results of the flatbed scanner by Del Guasta (2022), supporting a study of the modelled snow drift on the Antarctic plateau by Libois et al. (2014). Yet, a contribution of blowing snow cannot be excluded in the investigated ice fog events at Dome C also at lower wind speeds, especially due to the few observations of ice fog at wind speeds below 4 m/s (see supplementary material S2 for more details on the effect of the wind speed on the ice fog microphysics).".*

- It is stated in the paper (l259) that the diurnal cycle boundary layer dynamics only weakly affects the development of the fog. I may be wrong but Fig 5 suggests the contrary to me. The fog layer deepens in local morning associated with the growth of the boundary layer (convective activity, as seen by the vertically homogeneous temperature in panel e) and the optical thinning near local noon and sharp increase in the relative fraction of sublimating particles might suggest a fog sublimation through vertical downward transport of dry air from the top of the convective boundary layer.

*Response: We agree with the interpretation of referee #1 and changed line 258f. to read "Between 00:00 UTC on 26 November 2023 and 00:00 UTC on 27 November 2023 the diurnal cycle affects the development of the ice fog event. The fog layer increases in height at local morning, which can be linked to convective activity and is supported by the vertically homogeneous temperature profile. At local noon the ice fog layer becomes optically thinner with a sharp increase in the fraction of sublimating particles, which can be caused by vertical downward transport of dry air from the top of the convective boundary layer.".*

- At several places in the paper (especially in the discussion), the authors mention the homogeneous freezing process. However the distinction between homogeneous freezing of – relatively large – pre-existing supercooled liquid water (SLW) droplets at   -38°C and the homogeneous nucleation of ice through freezing of aerosol solution particles (at temperature below -38°C) should be distinguished. Vignon et al. 2022 suggested that the ice fog they observed was formes by homogeneous nucleation (freezing of small aerosol droplets), as the events they focused on took place in winter during which the temperature is too cold for SLW to exist at Dome C. Please distinguish the two processes during the analysis and clarify the text because it is confusing at some places and the interpretation might be not always correct.

*Response: We adapted all occurrences of homogeneous freezing to refer to homogeneous freezing of aqueous solution aerosol particles. Line 233 was changed to "This sudden, transient increase in ice particle complexity, visible by highly-speckled diffraction pattern images, can be a consequence of homogeneous freezing of solution aerosol particles, as homogeneously frozen ice particles recorded with SID-3 have been associated with increased surface roughness (Ulanowski et al., 2014; Schnaiter et al., 2016).". This is furthermore clarified in lines 400-405, which now read "The first pathway is homogeneous nucleation, where aqueous solution aerosol particles freeze spontaneously at temperatures below approximately -38 °C (Koop et al., 2000; Schneider et al., 2021). The exact temperature and relative humidity depends on the droplet size and water activity (Koop et al., 2000). In the following, by homogeneous freezing we refer to homogeneous freezing of aerosol solution particles and not to the freezing of larger pre-existing supercooled liquid water droplets.". Line 410-411 is adapted to "These findings support homogeneous freezing of aerosol solution particles as a possible nucleation pathway for locally formed ice fog events on the Antarctic plateau, as suggested by Vignon et al. (2022)".*

- This is a comment related to the previous one. I find really unfortunate that the authors do not show an analysis of a liquid fog event (which could possibly lead to homogeneous freezing of supercooled liquid water drops). They mention that this is left for a future work, but the current paper is not that long and can include an additional case study. Such an additionnal analysis would allow to strengthen the interpretation of the spherical habits in relation with relative humidity wrt liquid, and to reinforce the conclusions about the occurrence of homogeneous freezing (of liquid droplets) at Dome C and its role in ice fog formation in summer. I would even say that the paper cannot keep the current message regarding the occurrence of homogeneous freezing without an additional investigation of liquid fog. Please see Ricaud et al. 2025 (https://doi.org/10.1016/j.polar.2025.101256) for additional information of nocturnal liquid fog at Dome C.

*Response: This work focuses on the microphysical properties of boundary layer ice particles. Therefore, in our opinion, the analysis of supercooled liquid fog events is beyond this work's scope. We will publish an analysis of the supercooled fog events in a separate work which is currently in preparation and will focus on these short-time high-concentration nocturnal liquid fog events with possible homogeneous freezing of pre-existing supercooled droplets. We adapted the comment about homogeneous freezing of aqueous solution aerosol particles from the investigated ice fog events in line 233 to be more suggestive: "This sudden, transient increase in ice particle complexity, visible by highly-speckled diffraction pattern images, can be a consequence of homogeneous freezing of solution aerosol particles, as homogeneously frozen ice particles recorded with SID-3 have been associated with increased surface roughness (Ulanowski et al., 2014; Schnaiter et al., 2016).".*

- I overall find the interpretation of the small-scale optical complexity parameter k not easy to follow at many places in the paper. Can you provides details about the physical meaning of this parameter and help the reader interpret its evolution.

*Response: We added additional details about the physical meaning of the small-scale optical complexity parameter $k_e$ to the methods section of the manuscript at lines 85-88, which now read "The GLCM p describes how often pairs of pixels with the same gray level occur along a given direction. Each element $p(\Delta x, i, j)$ gives the total number of specific pairs of gray-level values i and j, which are separated by a pixel distance $\Delta x$. Different textual features can be derived from the GLCM. One is the energy $E(\Delta x)$, which is defined as the sum of the squared elements of the GLCM*

$$E(\Delta x) = \sum_{i=0}^{m-1} \sum_{j=0}^{m-1} p(\Delta x, i, j)^2 \qquad (1)$$

*with normalised co-occurrence matrix elements $p(\Delta x, i, j)$ for co-occurrence matrix size m (Lu et al., 2006). The normalised energy feature parameter $k_e$ is the coefficient of an exponential fit to $E(\Delta x)$ and was found to be the most robust parameter to measure small-scale optical complexity from the diffraction patterns (Schnaiter et al., 2016).". Furthermore, lines 230-234 are adapted to "This sudden, transient increase in ice particle complexity, visible by highly-speckled diffraction pattern images, can be a consequence of homogeneous freezing of solu-*

*tion aerosol particles, as homogeneously frozen ice particles recorded with SID-3 have been associated with increased surface roughness (Ulanowski et al., 2014; Schnaiter et al., 2016)." and lines 419-421 are adapted to "The larger amount of water vapour furthermore favours a higher growth speed, which causes more complex ice crystals with strongly speckled diffraction patterns to occur at Fairbanks in comparison to Dome C.".*

Minor comments :

l12-13 : This is a quite strong statement given the conclusions from this study only holds for a limited period of time at a single location in Antarctica.

*Response: While we measured the ice particle microphysical properties at a single location in Antarctica for a limited period, the cold, dry and clean conditions that lead to the observed lower particle concentration are characteristic for the Antarctic plateau in general (Dalrymple, 1966). Due to the limited temporal and spatial statistics of our observations we adapted lines 12-13 to be more suggestive "Our findings show that Antarctic boundary layer ice particles may need to be parametrised differently than their Arctic counterparts due to distinct conditions on the Antarctic plateau."*

l16 'a fraction of about 40 % clear-sky precipitation' : this number is one estimate from one particular location. The contribution of clear-sky precipitation to the overall Antarctic precipitation is still an open question.

*Response: It is now highlighted that this number is from one specific location and may not be representative for the Antarctic plateau in general. Lines 15-16 are updated to "Clear-sky precipitation (diamond dust) can be a significant contribution to the total precipitation on the Antarctic plateau with an observed fraction of 40 % of the total precipitation at Dome F (Dittmann et al., 2016)."*

L33 : 'in a commonly used parameterisation' : are you sure the Girard & Blanchet parameterisation is 'commonly used' ? To my knowledge, I am not aware of many studies leveraging their microphysical parameterisation.

*Response: Different studies on ice fog observations use or refer to this parametrisation for ice fog and diamond dust particle sizes and concentration (e.g. Schmitt et al., 2024b; Vignon et al., 2022; Zeng, 2018; Gultepe et al., 2017; Ricaud et al., 2017; Kuhn and Gultepe, 2016; Gultepe et al., 2015; Schmitt et al., 2013). As the phrasing 'commonly used' is imprecise, we removed it from lines 33-35: "A parametrisation of Arctic boundary layer ice particles by Girard and Blanchet (2001b) distinguishes ice fog with particle sizes below $30\,\mu m$ and concentrations above $1 \cdot 10^3\,L^{-1}$ from diamond dust particles with sizes larger than $30\,\mu m$ and concentrations below $4 \cdot 10^3\,L^{-1}$." and lines 9-11 "The mean particle concentration of the ice fog at Dome C ($3.9\,L^{-1}$) is found to be lower than in parametrisations of Arctic ice fog and lower than the concentration of anthropogenically*

*influenced urban ice fog measured at Fairbanks, Alaska during a three-year study
with the same instrument (90 L⁻¹)."*

L37 : 2.85 and 2.57 K ? Where do those numbers come from ? How can such
close numbers come from very different radiative forcings ? Girard & Blanchet
(2001b) quantify the reduction in cooling rate associated with diamond dust
and fog. The two types of cloud can lead to very different surface warming only
if integrating over different time lengths.

*Response: These numbers originate from table of 5 of Girard and Blanchet
(2001a). The similar values for the surface warming of ice fog and diamond
dust are due to the evolution of simulations of the two types of boundary layer
ice particle events. Multiple episodically appearing diamond dust occurrences
with higher peak radiative forcing are compared to one rather continuos three-
day ice fog event with a lower peak radiative forcing. This is now clarified in
lines 36-38: "During winter time falling diamond dust is simulated to locally
increase the downward infrared radiative flux by up to 60 W/m² in compari-
son to no low-level ice particle events. Integrating the longwave radiation of a
three-day quasi-continuous ice fog event and a three-day episodically occurring
diamond dust event, mean surface cooling rates of -2.85 K/day and -2.57 K/day,
respectively, were found (Girard and Blanchet, 2001a)."*

Figure 1 : 'Time of flight in a.u.' can you specify the meaning of this label in
figure's caption ?

*Response: A description is added to the figure caption: "The time of flight is
the time that the trigger intensity is above the threshold for particle detection.".*

L112 '10%' How has this number been estimated ?

*Response: At Dome C, stable oscillations occurred in the mass flow caused by
the pump. These oscillations are included in the uncertainty in mass flow of 0.5
normal liters per minute. Line 111-112 now reads "The relative uncertainty in
particle concentration is estimated 10 % due to the uncertainty in sample flow
dominated by a stable oscillation in mass flow caused by the used pump.".*

L130-132 : What should the reader conclude about the difference of mainte-
nance between the two periods ? Should we expect differences in data quality ?

*Response: There should not be differences in data quality during the operational
time in both periods. However, the downtimes increased in the second period
with only remote access due to occurrences of connection errors between the
computer and the instrument. Lines 131-133 have been updated to "The in-
strument was under remote operation with at least daily access during the first
three weeks and no access for the last four weeks of the deployment. The data
quality is comparable between the two periods but occasional downtimes occurred*

*during the no access period due to connection errors between the computer and the instrument.".*

L144 : The proper reference for this 7 m/s wind speed threshold to detect the occurrence of drifting snow is Libois et al. 2014 Libois, Q., Picard, G., Arnaud, L., Morin, S., and Brun, E.: Modeling the impact of snow drift on the decameter-scale variability of snow properties on the Antarctic Plateau, J. Geophys. Res.- Atmos., 119, 662–681, https://doi.org/10.1002/2014JD022361,2014.

*Response: The citation to Libois et al. (2014) was added. Line 144 is updated to "To have an increased particle count rate at the measurement height of PPD-2K the wind speed must exceed about 7 m/s as seen in the multi-year results of the flatbed scanner by Del Guasta (2022), supporting a study of the modelled snow drift on the Antarctic plateau by Libois et al. (2014)."*

L148 : around → round

*Response: Corrected.*

L149 aerosol → aerosols

*Response: Corrected.*

L163 : 'regularly occurs' Please refer to Genthon et al. 2017 .Genthon, C., Piard, L., Vignon, E., Madeleine, J.-B., Casado, M., and Gallée, H.: Atmospheric moisture supersaturation in the near-surface atmosphere at Dome C, Antarctic Plateau, Atmos. Chem. Phys., 17, 691–704, https://doi.org/10.5194/acp-17-691-2017, 2017.

*Response: The citation was added. Line 163-164 now read "This method avoids inaccurate measurement due to ice deposition on the sensor in an environment, where high supersaturation with respect to ice regularly occurs (Genthon et al., 2017)."*

L163 : for the calculation of saturation vapor pressure.

*Response: Added.*

L181 : The last sentence is awkward. Please reformulate.

*Response: Lines 179-182 are updated to "Fig. 3 shows the habit fractions determined with the machine learning convolutional neural network analysis of the diffraction patterns by Schmitt et al. (2024a) as functions of temperature, relative humidity, temperature inversion and particle size. It can be seen in Fig. 3a that the fraction of rough particles was below 10 % in all ranges of relative*

*humidity with respect to ice.".*

L185 and Figure 3c : an additional panel with the relative humidity wrt liquid in xaxis would be helpful.

*Response: The relative humidity with respect to liquid water was added to Fig. 3 (see figure below). The first sentence of the caption of Fig. 3 was changed to "Particle fractions determined with the machine learning convolutional neural network method (Schmitt et al., 2024b) for different ranges of relative humidity with respect to ice ($RH_{ice}$) **(a)**, relative humidity with respect to liquid water ($RH_{water}$) **(b)**, temperature inversion **(c)**, particle size **(d)** and automated weather station temperature ($T_{AWS}$) **(e)**.". Lines 193-198 now read "Fig. 3b shows that the fraction of spherical particles (droplets) increases with increasing relative humidity with respect to liquid water ($RH_{water}$). Furthermore, the fraction of pristine particle decreases with increasing $RH_{water}$." and at lines 205-211 "In Fig. 3e it can be noted that the fraction of pristine ice particles decreased and the fraction of sublimating particles increased for increasing temperature. This is in good agreement with previous findings that ice crystals tend to grow to more pristine shapes at colder temperatures (Bailey and Hallett, 2009; Schnaiter et al., 2016). A larger fraction of spherical particles was only found between -40 °C and -34 °C. This was the temperature range of two high concentration supercooled liquid fog events that occurred on 17 and 18 December 2023 and dominated the total recorded particles at these temperature ranges. A detailed analysis of these liquid fog events is beyond the scope of this study focusing on the ice particle microphysical properties. It will be reported in a future work.".*

[Figure]

Furthermore, Fig. B1 was changed accordingly (see figure below). The first sentence of the caption of Fig. B1 was adapted to "Particle fractions determined with the Fourier method (Vochezer et al., 2016) for different ranges of relative humidity with respect to ice ($RH_{ice}$) **(a)**, relative humidity with respect to liquid water ($RH_{water}$) **(b)**, temperature inversion **(c)**, particle size **(d)** and automated weather station temperature ($T_{AWS}$) **(e)**."

[Figure]

Figure 5f : please add the time series of RHliq as well. This might give insights into the ice nucleation mechanism at play.

*Response: The relative humidity with respect to liquid water was added to Fig. 5f (see figure below). The corresponding sentences in the figure caption is adapted to "Temporal evolution of the ice fog event of 25 November 2023 with lidar backscattering signal in a.u. **(a)** and lidar backscattering linear depolarisation δ **(b)** for the height h in m, particle size distribution **(c)** and mean particle concentration (N) over 600 s **(d)**. Instrument downtimes are shaded in red. **(e)** shows the temperature (T) measured at heights of 3 m and 42 m above ground in °C. Temperature inversions are highlighted. **(f)** shows the relative humidity with respect to ice and to liquid water (RH_ice,RH_water) at heights of 3 m and 42 m.".*

[Figure]

Sect. 3.2 Can you explain why you chose this particular event and not another. Additional motivation and justification are needed.

*Response: This particular event was chosen because it was the ice fog event with the longest duration observed during the measurement period at Dome C. Lines 294-296 now read "This event is the ice fog event with the longest duration observed during the measurement period at Dome C. Therefore, it is highlighted in this work."*

L210 : 'radiative cooling' : not necessarily. Vignon et al. 2022 show that ice fog formation can be triggered through local cooling of the air associated with turbulent mixing.

*Response: Line 210 was updated to "This confirms that the boundary layer ice crystals were locally formed on the Antarctic plateau.".*

L254 : ' layer started to weaken' → ice fog layer depth started to decrease ?

*Response: Line 254 is clarified and now reads "Starting at 18:00 UTC on 26 November 2023 the lidar backscattering signal of the ice fog layer starts to de-*

*creases again with a simultaneous decrease in particle size and in the fraction of pristine ice particles."*

L291 : Can you really qualify a fog event as a 'precipitation event' ?

*Response: The wording was changed from "precipitation events" to "boundary layer ice particle events" in line 291 and additionally from "precipitation event" to "boundary layer ice particle event" in line 349.*

L294 : please recall the considered time period here.

*Response: The considered time period is added to line 294: "In this section, the microphysical properties such as particle habit, concentration, size distribution and small-scale optical complexity ($k_e$) of ice particles measured with PPD-2K during the full measurement period between 21 November 2023 to 5 January 2024 are compared for ice fog events (labelled: if) and cpdd events at Dome C, Antarctica.".*

L302 : 'measurement times are excluded when pollution ...' this is a repetition from the Methods section.

*Response: The sentence "Measurement times are excluded when pollution from Concordia station was possible due to the wind direction." was removed in line 302-303.*

L344 : emission → emissions

*Response: Corrected.*

Appendix : There are too many appendices for a quite short paper (4). In particular, Appendix A and D seem not absolutely critical and can be summarized in a few words in the main text.

*Response: Appendix A is summarised in line 210-212 "Furthermore, aerosol pollution can also be excluded due to the southerly winds during the event, which transported exhaust emissions from generators and vehicles operated at Concordia station away from the measurement site." Appendix D is summarized in line 131-133 "The instrument was under remote operation with at least daily access during the first three weeks and no access for the last four weeks of the deployment. Occasional downtimes occurred during the no access period due to connection errors between the computer and the instrument.". Both figures are moved to the supplementary material.*

**Authors' Responses to Anonymous Referee #2**

Review of "Unique Microphysical Properties of Small Boundary Layer Ice Particles under Pristine Conditions on Dome C, Antarctica" by Adrian Hamel et al.

Overview of the paper :

This paper analyses ground-based, optical in situ measurements of falling ice crystals sampled by the PPD-2K during several weeks in Summer at Dome C. Number concentrations, spherical equivalent diameters, habits and morphological complexity are derived from the optical measurements using well-established algorithms and methods. These microphysical properties of ice crystals with size larger than 11 µm are statistically analyzed for different thermodynamic conditions measured at the station. A lidar is also used to identify specific "meteorological" events such as precipitation, diamond dust and ice (or liquid) fogs. The results of the paper begin with an interesting overview of the ice crystal habit fraction for different atmospheric conditions and hydrometeor sizes. The paper then focuses on a case study encompassing ice crystal precipitation and an ice fog event. The temporal evolution of the microphysical properties of the ice fog is presented clearly. The analysis shows that there are distinct properties between ice crystal precipition and ice fogs, which can be captured by the modes of the particle size distribution (PSD) measured by the PPD-2K. A statistical analysis is performed on the microphysical properties of almost all ice fog and cirrus precipitation or diamond dust events. The PSD, concentration and ice crystal morphology parameters are inter-compared and also compared to previous measurements carried out in Fairbanks. The differences in these properties are discussed as are the possible nucleation processes occurring in ice fogs. The paper concludes whith a discussion of the specific properties of ice fogs in inland Antarctica, emphasizing the necessity of new parametrization of ice crystal-related processes and properties in this region.

The unique instrumental setup deployed at Dome-C for this study allows for detailed and pertinent characterization of the microphysical properties of ice crystals. In my opinion, the strength of the paper lies in its analysis of the morphological properties (habit and complexity) of ice crystals for different thermodynamic conditions and "cloud types". This approach is essential for improving our understanding and modeling of the microphysical processes that control the life cycle of ice crystals and precipitation in this region of the globe. The relevance of this study is clear, and the results are original and very interesting. While the description of the methods and results is of good quality, it could be improved by avoiding certain repetitions and providing clarifications, mainly on the sampling strategy/limitations and some of the results. I would recommend minor revisions before the manuscript can be considered for publication in ACP.

*Response: We are grateful to Anonymous Referee #2 for the positive assessment and valuable suggestions to improve the manuscript. Below, we detailed*

*responses to the comments are given and corresponding revisions are detailed.*

General comments :

1. Introduction/motivation : I think the motivation/storyline of this study could be improved to deliver a clearer message. It should state more clearly what is needed in models to better simulate the life cycle of ice fogs. Modeling and characterizing the properties of ice precipitation is also subject to significant uncertainty in Antarctica. For instance, I think there is a need to improve the modeling of the fall speed of ice crystals which depends on their size and shape. The authors also mention that their measurements are valuable for validation of satellite retrievals. I would suggest they clarify their statement by demonstrating how their measurements could improve the remote sensing retrievals (not just satellite also more importantly ground based lidar/radar retrievals). Strong assumptions are needed to retrieve basic properties such as ice number concentration and reliable retrievals are even more challenging near the ground, especially in Antarctica. Did the authors have in mind that their measurements could be used to compute look-up tables or parametrization relating ice water content to radar reflectivity or lidar backscattering for several ice crystal morphologies ? (please refer to the detailed comments below)

*Response: It is now specified in the motivation that the measurements of ice particle concentration and size can be used for the validation of ground, airborne and spaceborne lidar and radar remote sensing instrumentation retrievals such as look-up tables or parametrisations. The possible application of the data to the retrieval of ice water content and the modeling ice particle fall speeds is additionally added. Lines 46-54 now read "Furthermore, the in situ measurements of microphysical properties of low-level atmospheric ice crystals are valuable for the validation of particle size and concentration retrievals from ground, airborne and spaceborne remote sensing systems, which currently depend on a small number of in situ measurements (Palm et al., 2011). For example, they can be used to improve look-up tables for ice particle size and concentration based on radar reflectivity or lidar backscattering (e.g. Aubry et al., 2024; Di Natale et al., 2022; Sourdeval et al., 2018). The enhanced knowledge about the ice particle microphysical properties like particle size, shape and concentration can improve the atmospheric modelling of inland Antarctica —for example modelling of ice water content (Aubry et al., 2024; Di Natale et al., 2022), ice particle fall speeds (Mishra et al., 2014) or radiative forcing (Girard and Blanchet, 2001a)—.".*

2. Sampling of ice crystals (Methods and results section) : I recognize the challenges involved in performing microphysical measurements at Dome C. Although the methods section is well described, i think some details are missing regarding the ice crystal sampling and the potential additional sizing and counting uncertainties arising from it. In particular, the impact of the pumping speed, the influence of wind speed and direction and underlying non isokinetic issues inside the instrument inlet should be addressed.

*Response: We added a discussion of the effects of changing wind speed to the ice particle sampling characteristics. An additional figure about the effect of the wind speed on the measured ice particle microphysical properties is added to the manuscript as supplementary material. The replies to the specific comments are given below.*

3. Liquid fog event (results section and discussion) : it is mentioned several times that a liquid fog was sampled during the campaign. The corresponding data are included in the "all event" dataset and seem to have a significant impact on the maximum measured particle concentration presented in Table 2 as well as on the PSD and habit fractions shown in Fig. 7.I understand that the authors intend to present this event in a separate publication. However, to clarify some of the results, I would suggest not including the liquid fog event in the "all event" category. Fig7a shows a sharp increase in the concentration for small sizes, which could be attributed to the presence of supercooled water droplets representative of this liquid fog event only. The habit fractions also appearto be impacted by this event which makes the inter comparisons less meaningful. Additionally, the wind speed and direction could also affect the sampling of small water droplets and larger ice crystals differently. I would recommend either dismissing the liquid fog event from the analysis (in the all category) or investigating this event separately (by adding a new category to table 2 and Fig. 7). This analysis could also provide information on the nucleation mechanisms in ice fogs.

*Response: We agree that it is beneficial to separate the liquid fogs from the 'all' dataset because this paper focuses on the microphysical properties of ice particles. Therefore, Fig. 7 (see figure below) and table 2 (see table below) are changed accordingly. 'all' now refers to the complete measurement period excluding supercooled liquid fog. For more details we refer to the replies of the specific comments below.*

[Figure]

| | Maximum measured particle concentration | | | |
|---|---|---|---|---|
| Averaging period | if | cpdd | all | FB |
| 600 s | $5.1 \cdot 10^1 \, \mathrm{L}^{-1}$ | $2.7 \cdot 10^1 \, \mathrm{L}^{-1}$ | $1.1 \cdot 10^2 \, \mathrm{L}^{-1}$ | $9.8 \cdot 10^3 \, \mathrm{L}^{-1}$ |
| 60 s | $6.0 \cdot 10^1 \, \mathrm{L}^{-1}$ | $4.2 \cdot 10^1 \, \mathrm{L}^{-1}$ | $6.9 \cdot 10^2 \, \mathrm{L}^{-1}$ | $2.5 \cdot 10^4 \, \mathrm{L}^{-1}$ |

Specific comments :

1. Introduction

Line 16 : Could be more specific about your statement that "a fraction of about 40% clear-sky precipitation" ? What do the authors mean by clear-sky precipitation : diamond dust only ?

*Response: By clear sky precipitation we refer to diamond dust, which is now clarified in lines 16-17 "Clear-sky precipitation (diamond dust) can be a significant contribution to the total precipitation on the Antarctic plateau with an observed fraction of 40 % of the total precipitation at Dome F (Dittmann et al., 2016)."*

Line 16-17 : In Walden et al., 2003 it is said that falling ice crystals collected on a gridded glass slide at the South Pole Station were photographed through a microscope. They showed that the effective radius of diamond dust was 12-15

µm ; 11 µm for blowing snow and 24µm for snow grains. These values seem quite small as the effective size is not always representative of the actual size of ice crystals, especially for hexagonal columns or plates. Walden et al., 2003 also show that diamond dust sizes range from 2 µm to 1000 µm. It think that it would be wise to mention the variability in ice crystal size or to specify that this 10 µm value correspond to an effective size. Since the PPD-2K cannot in principle detect crystals small than 10 µm, readers may wonder how these new measurements will contribute to our understanding of microphysical properties when most of the crystals are smaller than 10 µm.

*Response: We changed the phrase to emphasis the large range of ground level atmospheric ice particle sizes on the Antarctic plateau. Lines 16-17 now read "Atmospheric ice crystals in this cold and dry place occur in a large size range between 2 µm and 1000 µm (Walden et al., 2003). In situ measurements of atmospheric ice particles remain scarce in this remote region, where challenging conditions only allow access in the few summer months of the year.".*

Line 20-22 : The authors mention that the clean environment of Antarctica enables us to study cirrus formation and that several measurements have been performed in the Antarctic boundary layer. This is confusing as cirrus clouds are typically located in the high/free troposphere. I would recommend to use the term ice clouds or ice containing clouds.

*Response: The phrase in lines 20-22 is updated to refer to ice clouds and now reads "It enables us to study ice cloud formation in a surface atmospheric environment that can be considered one of the cleanest on the planet (Heumann, 1993)."*

Lines 24-29 : The authors state that previous instruments could not reliably determine the properties of ice crystals with sizes below 50 µm. While I agree with this, it is not consistent with what is written in lines 16-17. Furthermore, i would recommend that the authors elaborate on why this information on 50µm ice crystal is necessary for improving the estimation of the radiative impact in the boundary layer, providing some references.

*Response: Lines 16-17 have been updated and now emphasise the size range that can be measured with PPD-2K: "Atmospheric ice crystals in this cold and dry place occur in a large size range between 2 µm and 1000 µm (Walden et al., 2003).". The microphysical properties influence the radiative properties of ice clouds. With a large fraction of ice particles in the size range below 50 µm on the Antarctic plateau (Walden et al., 2003), their microphysical properties are important for the radiative forcing. Lines 21-29 now read "[...] they cannot reliably determine ice crystal concentration, shape, complexity and size distribution at a high temporal resolution for particles with sizes below 50 µm (Kaye et al., 2008; Ulanowski et al., 2010; Gultepe et al., 2017). Scanning electron microscopy of Formvar replica can obtain shape and size information in this*

*size range. Nevertheless, it is currently only available for some tens of snap-shots covering a measurement time of 30 s each (Santachiara et al., 2016). The microphysical properties of the boundary layer ice particles need to be known to accurately determine the cloud radiative impact (Shupe and Intrieri, 2004; Cox et al., 2019). The large fraction of ice particles at sizes below 50 μm on the Antarctic plateau (Walden et al., 2003) emphasises the importance to properly parametrise the microphysical properties of ice particles in this size range in radiative transfer models.".*

Line 35 : Is the parametrization used in Girard and Blanchet, 2001a, the only one available and most commonly used ? Are there any other parametrization with lower number concentrations ?

*Response: Many publications that measure ice fog characteristics refer to this publication as a way to distinguish ice fog and diamond dust (e.g. Schmitt et al., 2024b; Vignon et al., 2022; Zeng, 2018; Gultepe et al., 2017; Ricaud et al., 2017; Kuhn and Gultepe, 2016; Gultepe et al., 2015; Schmitt et al., 2013). The parametrisation is generally in good agreement with Arctic ice fog observations (e.g. Gultepe et al., 2017, 2015). To our knowledge there are no common parametrisations that give lower ice fog particle concentrations.*

Line 36: the sentence beginning with "During winter time falling diamond dust is simulated to increase the downward infrared flux by up to ....." is unclear. Increased relative to what ? Not taking diamond dust into account ?

*Response: We clarified the phrase in lines 36-37, which compares the long-wave downward radiation of diamond dust and ice fog events to no low-level ice particles: "During winter time falling diamond dust is simulated to increase the downward infrared radiative flux by up to 60 W/m² and ice fog of about 7.4 W/m² in comparison to no low-level ice particles."*

Line 42 : What is the open question regarding particle morphology ?

*Response: Information about the particle morphology especially at high temporal evolution is scarce for particle sizes below about 50 μm on the Antarctic plateau. Lines 41-43 are adapted to "The single particle counter and diffraction pattern imager PPD-2K (Kaye et al., 2008) can add important information about the particle morphology at a high temporal resolution and thus about the radiative properties and the nucleation process in locally formed Antarctic ice fog."*

Line 47 : it is stated that in situ measurements of the microphysical properties of ice crystals are valuable for validating of satellite data retrievals and atmospheric modelling. This is quite a general justification. I would recommend that the authors be more precise by indicating some of the shortcomings of remote sensing retrievals (not only satellite as i think your results might be more interesting for radar/lidar retrievals from ground-based systems). Regarding atmospheric modeling it would also be interesting to show which parametrization or process should be improved.

*Response: We agree that the measurements can also be used to validate ground based remote sensing systems. In addition, advantage of our findings for atmospheric modelling were specified to the improved knowledge of the ground level particle size distribution and particle concentration. Please see the response to general comment one for more details.*

2. Methods

I would reorganize this section slightly, starting with a short paragraph mentioning the sampling procedure, and the instruments and tools that will be used in the study. The atmospheric instrumentation could come first but the authors need to explain why the lidar measurements and air parcel back trajectories are necessary. The sampling strategy could also be developed further, with a focus on the possible impact of wind on the measurements.

*Response: An introductory paragraph was added to the methods section to better organise the section. It explains why the lidar measurements are necessary. Lines 72-55 now read "This section provides details about the instrumentation used for the characterisation of the microphysical properties of the boundary layer ice particles on the Antarctic plateau. First, the single particle counter is introduced, which provides the in situ ice particle observations, and the setup at Dome C is detailed. Next, the lidar instrument used to identify different events of boundary layer ice particles is described. Furthermore, the atmospheric instrumentation is detailed, which measures the atmospheric conditions like temperature, relative humidity, wind speed and wind direction.". Furthermore, an explanation why the back-trajectories are needed is added in lines 169-170 "These back-trajectories are used to verify that the events of boundary layer ice particles originate from the Antarctic plateau and do not have maritime influences.". Additional details about the sampling methods and discussions about the influence on the wind are added. For the detailed changed please see the replies to the comments below.*

Lines 96-97 : Could you explain why and how the diffraction patterns with Mie fringes for spherical particles would increase the mean complexity ? Could you please elaborate on this ?

*Response: The increase in $k_e$ for spherical particles is an observational finding (see figure below) likely caused by the high number of neighbouring pixels with contrasting intensity, which are inherent for diffraction patterns of spherical particles showing concentric rings. Lines 96-97 are adapted to "In this work, spherical particles are not considered for the calculation of the mean of $k_e$ because $k_e$ is only well-defined for solid surfaces. Furthermore, the diffraction patterns with Mie fringes could falsely increase the mean complexity due to*

the high number of neighbouring pixels with contrasting intensity inherent for diffraction patterns consisting of concentric rings."

[Figure]

Figure 1: The $k_e$ values for the different particle habits determined by the two classification methods. It can be seen that spherical particle shapes lead to high $k_e$ values (ir: irregular; lr: large; pr: pristine; rg: rough; sr: small rough; sm: small; sp: spherical; sb: sublimating).

The figure is added to the supplementary material.

Line 111-112 : How was this uncertainty estimated ? Is a mass flow of 5 l /min sufficient to achieve accurate measurements ? In this respect, did the authors investigated the impact of the flow rate on the measured concentrations and on the size of ice crystals ? What is the sampling speed in the instrument's inlet ? I'm wondering if you are working in isokinetic conditions ?

Response: The uncertainty is due to stable oscillations in the mass flow caused by the pump. These oscillations are included in the uncertainty in mass flow of 0.5 normal liters per minute. Line 111-112 now reads "The relative uncertainty in particle concentration is 10 % due to the uncertainty in sample flow dominated by a stable oscillation in mass flow caused by the used pump.". The mass flow of 5 normal liters per minutes has been chosen to be consistent to previous measurements and has been proved to yield good results (e.g. Schnaiter et al., 2016). The investigation of the effect of the sampling speed on the concentration and size was not investigated at Dome C. The sampling speed at the top of the inlet is about 0.1 m/s and it is non isokinetic due to changing wind speed at the measurement site. For effects of the wind speed please see the reply to the comment about line 140-145).

Line 132 : This sentence is unclear. Are the inlet and the metal tube the same thing? Could you please clarify this ?

*Response: The inlet is a horn-shaped inlet, which is mounted on top of the metal tube. To clarify this, lines 132-134 now read: "The inlet is situated approximately 6 m above ground level with an upward facing horn-shaped nozzle (diameter: 39 mm) to minimize sampling artefacts. The horn-shaped nozzle is connected to the PPD-2K with a 50 cm long metal tube with an inner diameter of 10 mm in a vertical straight line".*

Line 140-145 : The author mention that, for low wind speeds, the data is usually excluded from the analysis due to aerosol pollution. However, what is the effect of wind speed and direction on the ice crystal sampling ? Do you expect there to be a lot of undersampling of falling ice crystals when the wind speed is high ?

*Response: The inlet cone is symmetric around the vertical axis, therefore no effects of the wind direction on the sampling are expected. Due to the particle inertia, an undersampling is possible at high wind speeds, as observed for aerosol particles in a similar size range to our observations and different inlets (Vanderpool et al., 2018). The possibility of undersampling at higher wind speeds is added to the description of the instrument setup in lines 145-147: "At high wind speeds the collection efficiency decreases with increasing particle sizes due to increasing particle inertia. While we did not observe this in the particle size distributions, we cannot exclude undersampling at high wind speeds due to decreasing collection efficiency.". We additionally added this possible bias in the discussion section of the manuscript in lines 397-401: "The wind speed at the measurement site can influence the observations. At high wind speeds blowing snow can increase the observed particle concentrations. Simultaneously, the collection efficiency likely decreases for the larger particle size at high wind speeds due to increasing particle inertia (Vanderpool et al., 2018). A figure of the particle concentration and particle habit fractions as a function of the wind speed are added as supplementary material (Fig. S2).".*

Figure 1 : What is the meaning of the color bar "time of flight" ?

*Response: A description is added to the figure caption: "The time of flight is the time that the trigger intensity is above the threshold for particle detection.".*

3. Results
Line 181: Could you clarify this sentence : "the fraction of rough particle was with 21% the highest" ?

*Response: Line 181 is rephrased and now reads "In Fig. 3a it can be seen that in all ranges of relative humidity with respect to ice the fraction of rough particles was below 10 %.". The absolute numbers changed due to the fixed pollution time*

*flagging (see other changes).*

Line 190 : Do you mean that sublimating ice particles could be frozen droplets ? This is interesting. Do you have a method such as the analysis of lidar depolarization or PPD-2k to distinguish between these two types of hydrometeors ?

*Response: Frozen droplets have similar ellipse-shaped diffraction patterns as sublimating particles. Therefore, particles classified as sublimating can be frozen droplets. For more information see Järvinen et al. (2016). To my knowledge there is no certain way to certainly distinguish frozen droplets from sublimating particles either from lidar or PPD-2K data unless the complete temporal evolution was captured and therefore the origin (liquid or frozen) of the particle is known.*

Line 202-203 : So is it plausible that the sublimating particles are frozen or quasi spherical droplets ? What level of confidence do you have in discriminating between the two types ?

*Response: We do not fully comprehend the term 'quasi spherical droplet'. Either a particle is frozen, then it can be non-spherical or quasi-spherical, or a particle is liquid and then it is spherical due to the surface tension at particle sizes in the order of ten microns. The machine learning classification (Schmitt et al., 2024a) is designed to be sensitive on asphericity in the concentric rings on the diffraction patterns. Therefore, there is a high confidence that a particle that is classified spherical is liquid. On the other hand, if a particle is classified sublimating there is chance for misclassification of a liquid particle. To quantify this, 100 random diffraction patterns of the spherical and sublimating class were manually validated. Out of 100 spherical particles no falsely classified sublimating were found while out of 100 sublimating diffraction patterns 54 falsely classified sphericals were found. Lines 202-209 have been updated to "The fraction of pristine particles increased with increasing particle size from about 5 % at particle sizes of about $10\,\mu$m to 56 % at particle sizes of about $130\,\mu$m. In addition, it can be noted that spherical particles almost only occurred at sizes below $20\,\mu$m and sublimating particles only occurred at sizes below $40\,\mu$m. The level of confidence of the separation between spherical and sublimating particles was tested with a manual analysis of 100 random diffraction patterns. While all of the 100 spherical diffraction patterns were confirmed spherical by manual analysis, 54 out of the 100 sublimating particles had spherical concentric rings on the diffraction patterns. This explains the higher fraction of spherical particles in the Fourier classification in Fig. 7 in comparison to the fraction of spherical particles in the machine learning classification. The descripency to the confusion matrix in Schmitt et al. (2024a) is likely caused by smaller sizes of spherical particles at Dome C leading to fewer concentric rings on the diffraction patterns in comparison to the training data set from Fairbanks.".*

Line 207 : I think it's a bit disturbing to call the ice fog a "ground level thin

cirrus"

*Response: Line 207 was changed to "ground-level, thin ice cloud". Furthermore, "cirrus ice fog events" in line 7 was changed to "ice fog events".*

Line 233 : How can you be sure that an increase in the ice crystal complexity is indicative of homogeneous freezing ? What about other mechanisms that could lead to an increase in complexity such as sublimation, riming, rapid growth . . . . ?

*Response: We agree that an increase in ice crystal complexity can also have other reasons than homogeneous freezing. Therefore, line 233 was adapted to be more suggestive "This sudden, transient increase in ice particle complexity, visible by highly-speckled diffraction pattern images, can be a consequence of homogeneous freezing of solution aerosol particles, as homogeneously frozen ice particles recorded with SID-3 have been associated with increased surface roughness (Ulanowski et al., 2014; Schnaiter et al., 2016).".*

Line 285 : Is there a way to discriminate between cirrus precipitation and diamond dust ? Their formation processes should be different as well as their microphysical properties.

*Response: The separation is difficult because thin, scattered cirrus clouds were present during the majority of the diamond dust events. A larger dataset would be needed to analyse diamond dust events, when no thin, scattered cirrus clouds are simultaneously present. As cirrus seeding cannot be excluded when there are scattered cirrus present (see Del Guasta et al. (2024)), the class combines "cirrus precipitation and diamond dust". The classification is equivalent to the class of "diamond dust" in Lawson et al. (2006), which also includes events when scattered cirrus clouds were present. Lines 285-287 are updated to "Cirrus precipitation and diamond dust events are combined because the times of clear sky precipitation could rarely be clearly separated from cirrus fall streaks. Usually, a combination of both events occurred due to the predominance of scattered, thin cirrus clouds. The cpdd class is similar to the class of diamond dust in Lawson et al. (2006) which also compromises events when thin, scattered cirrus clouds were present.".*

Line 306-308 : Can you really be sure that all events exhibit the ice fog mode at 11 µm given that the instrument cannot quantify ice crystals smaller than 11 µm ? It is also possible that the counting is overestimated in this size range. Is there typically a larger measurement uncertainty in this size range ?

*Response: There is no known overestimation or larger measurement uncertainty in the smaller size ranges observed in previous deployments of PPD-2K. Under the assumption that there was no size mode towards the smaller size detection limit of PPD-2K in all events at Dome C, the overestimation would need to be*

*multiple orders of magnitude. This overestimation would have been identified in previous cloud chamber measurements with PPD-2K Hamel et al. (2025). Furthermore, the particle size distribution of ice fog in Fairbanks Alaska measured with PPD-2K by Schmitt et al. (2024b) is in good agreement with the particle size distribution of Fairbanks ice fog measured with a Formvar replicator by Schmitt et al. (2013).*

Line 331 and Table 2 : The concentration of droplets in liquid fog is included in the analysis to derive a maximum concentration during the complete measurement period. I think that it would be more relevant to separate these events from the other cases and even more pertinent to present the PSD of this liquid fog event.

*Response: We agree with referee 2 that it is advantageous to remove the supercooled liquid fog events from the 'all' class. Fig. 7 and table 2 have been updated accordingly. We decided not to show the microphysical properties of the supercooled liquid fog droplets in the manuscript because this would be beyond the scope of this work focused on ice particle microphysical properties.*

- *The first sentence of the caption of Fig. 7 is updated to "Ice particle microphysical properties for periods with low-level ice clouds (labelled if for ice fog), for periods with high-level streaks of ice crystals (labelled cpdd for cirrus precipitation - diamond dust) and for the complete measurement period (excluding supercooled liquid fog events) between 21 November 2023 and 09 January 2024 (labelled: all).".*

- *The first sentence of the caption of table 2 is updated to "Maximum measured particle concentration of particles larger than $11\,\mu$m for time averaging of $60\,s$ and $600\,s$ during the ice fog (if) events, cirrus precipitation and diamond dust events (cpdd) and the complete measurement period excluding supercooled liquid fog events (all) at Dome C.".*

- *Lines 301-302 are changed to "The complete measurement period from 21 November 2023 to 09 January 2024 excluding supercooled fog events is added (labelled: all).", lines 322-324 are changed to "The mean particle concentration of the complete measurement period excluding pollution times and supercooled liquid fog events is with $0.9\,L^{-1}$ lower than during the cpdd and ice fog events.".*

- *Lines 327-332 are changed to "The maximum concentration during the complete Antarctic data set (excluding supercooled liquid fog events) is $6.9 \cdot 10^2\,L^{-1}$ over $60\,s$. It is higher than the maximum particle concentrations during the lidar classified cpdd and ice fog events with maximum concentrations of $4.2 \cdot 10^1\,L^{-1}$ and $6.0 \cdot 10^1\,L^{-1}$ over $60\,s$. The high peaks in particle concentration outside the cpdd and ice fog events occurred only for periods of a few minutes. Therefore, we cannot exclude that these concentration peaks were caused by influences from maintenance of the measurement shelter."*

- *Lines 339-342 are changed to "Of the complete measurement period 57 % had irregular, 34 % had columnar, 7 % had plate-like and 2 % had spherical diffraction patterns.".*

- *Lines 347-348 are changed to "The mean small-scale complexity of the Dome C ice fog events, cpdd events and the complete measurement period at Dome C is similar with means of 4.21, 4.24 and 4.22, respectively.".*

Why did the author choose to consider liquid fogs in a separate paper ?

*Response: The liquid fogs are beyond the scope of this work which focuses on the microphysical properties of boundary layer ice particles on the Antarctic plateau. Details about the supercooled liquid fogs will be reported in a future study. In line 204-206 it is now added "A detailed analysis of these liquid fog events is beyond the scope of this study focusing on the ice particle microphysical properties. It will be reported in a future work."*

Line 332 : I think the comments on ice fog from Fairbanks have already been mention earlier.

*Response: The repetition in line 332 has been removed.*

Line 337-340 : It is indeed surprising that the habit fractions are so similar. Are the thermodynamic conditions similar too ? How would you explain this ? Once again, the low level supercooled liquid clouds should be excluded from this analysis and assigned to a specific liquid fog category (in table 2 and Fig 7a).

*Response: The growth conditions can differ strongly in temperature and relative humidity for ice particles that grow near ground level in comparison to ice particle that sedimented from a few hundred meters altitude (Del Guasta et al., 2024; Ricaud et al., 2017). While there are only relatively small differences seen in the Fourier analysis, the particle habits of the machine learning analysis differ more. Therefore, the similar habit fractions in the Fourier analysis method may be caused by a lower sensitivity in comparison to the machine learning method. An additional comment is added in lines 338-442 "This is an unexpected finding because the ice fog and cpdd particles have different growth regions and thus different growth conditions (Del Guasta et al., 2024; Ricaud et al., 2017). It is possible that the Fourier method is not as sensitive on the consequential differences in particle shapes as the machine learning method, because distinct differences between the habits of the cpdd and ice fog events are seen with the machine learning method in Fig. 7e."*

Line 357 : This sentence should be rewritten as "is with" is unclear. During cpdd events, 31% of ice crystals are pristine crystals .....

*Response: Lines 356-357 are rephrases and now read "During the cpdd events at Dome C, the fraction of pristine particles is 31 %. This fraction is higher than the fraction of pristine ice particles during the Dome C ice fog events (17 %) and during ice fog at Fairbanks (15 %).".*

Line 363 : This sentence could be placed in the discussion section, as it has already been mentioned several times in the results section.

*Response: We removed the repetition in line 363 because it is already mentioned in the discussion in lines 415-424. Lines 361-363 now read "Furthermore, the ice fog at Fairbanks, Alaska has a high fraction of 43 % rough particles in comparison to the Antarctic ice fog events with a low fraction of 2 % rough particles.".*

Line 376 : The maximum dimension can be 2.5 times the spherical equivalent diameter. I suggest placing this information the methods section or result section when presenting PPD-2K measurements.

*Response: The sentence is removed in line 376 because it is a repetition of lines 107-109.*

Discussion : This section should include a brief description of the limitations of the measurements due to sampling and possible contamination from blowing snow, and how it could be improved.

*Response: A brief description of the limitations of the measurements due to sampling and possible contamination from blowing snow was added to the discussion section of the manuscript in lines 397-401: "The wind speed at the measurement site can influence the observations. At high wind speeds blowing snow can increase the observed particle concentrations. Simultaneously, the collection efficiency likely decreases for the larger particle size at high wind speeds due to increasing particle inertia (Vanderpool et al., 2018). A figure of the particle concentration and particle habit fractions as a function of the wind speed are added as supplementary material (Fig. S2)."*

Appendix : Appendix A and D could be summarized in the main text.

*Response: Appendix A and D have been summarized in the text and the figures are added to the supplementary material. Please see the answer to the same comment by referee #1 for details.*

**Other changes**

*μ has been better formatted in figure labels.*

*An error was found in the calculation of the mean particle concentration of the comparison dataset from Fairbanks, because the dataset uses a dynamic, non-constant integration period, which was originally not taken into account in the calculation of the mean concentration. This is fixed now and the mean particle concentration of the Fairbanks dataset has been corrected to $9.0 \cdot 10^1 \, L^{-1}$ (see lines 11, 325, 416 and 442). This quantitative change does not alter the interpretation of the measurement data.*

*A minor error in the removal of the measurement times with possible pollution from Concordia station was found. The aerosol measurements of Virkkula et al. (2022) took place at the ATMOS shelter at Dome C, which has a 10° lower angle to Concorida station's generators in comparison to the PHYSICS shelter where PPD-2K was placed. This is corrected in*

- *Table 1*

- *Lines 141-144 "For wind speeds $1 \, m/s < v < 2 \, m/s$ wind directions between 350° and 140° are excluded and for wind speed $v > 2 \, m/s$ wind directions between 10° and 100° are excluded."*

- *A new version was uploaded to the data repository with the improved pollution flagging. This is adapted in lines 449-450 "PPD-2K measurement data from Dome C, the depolarisation lidar data and the measurement tower temperature and humidity data are available on https://doi.org/10.5281/zenodo.17616458 (version v3)."*

*Consequently, the times where pollution was possible were different which lead to minor changes in the results which are detailed below:*

- *Lines 181-184 now read "In Fig. 3a it can be seen that in all ranges of relative humidity with respect to ice the fraction of rough particles was below 10 %."*

- *Lines 203-204 now read "Fig. 3d shows the particle fractions depending on particle size. The fraction of pristine particles increased with increasing particle size from about 5 % at particle sizes of about 10 $\mu$m to 61 % at particle sizes of about 130 $\mu$m."*

- *Lines 297-301 now read "Out of the non-polluted operational periods 11.9 % are classified as cpdd events and 11.1 % are classified as ice fog events. The rest is prominently clear sky and occasional liquid cloud cover. Two events were observed when ice fog and cirrus precipitation occurred simultaneously (2.7 % of the non-polluted operational period)."*

- *Lines 319-323 now read "The mean particle concentration during the ice fog events at Dome C is with $3.9 \, L^{-1}$ about four times higher than during the cpdd events at Dome C with $9.1 \cdot 10^{-1} \, L^{-1}$. Ice fog is known for its higher particle concentration in comparison to diamond dust events (Girard and Blanchet, 2001b). The mean particle concentration of the*

*complete measurement period excluding pollution times and supercooled liquid fog events is with $0.9\,L^{-1}$ lower than during the cpdd and ice fog events.".*

- *Table 2*

- *Lines 328-329 now read "It is higher than the maximum particle concentrations during the lidar classified cpdd and ice fog event with maximum concentrations of $4.2 \cdot 10^1\,L^{-1}$ and $6.0 \cdot 10^1\,L^{-1}$ over $60\,s$.".*

- *Lines 347-348 now read "The mean small-scale complexity of the Dome C ice fog events, cpdd events and the complete measurement period at Dome C is similar with means of 4.21, 4.22 and 4.22, respectively.".*

- *Lines 355-356 now read "19\,% of the ice fog particles from Dome C and 16\,% of the ice fog particles from Fairbanks, Alaska have sublimating diffraction patterns, while the cpdd events at Dome C have a lower fraction of sublimating particles of 6\,%.".*

- *Lines 376-377 now read "The cpdd events exhibit a mean particle concentration of $9.1 \cdot 10^{-1}\,L^{-1}$, which is within the expected range of concentrations lower than $4 \cdot 10^3\,L^{-1}$ by Girard and Blanchet (2001b).".*

- *Line 383 now reads "During ice fog events we observed a mean concentration of $3.9\,L^{-1}$.".*

- *Lines now read "The mean particle concentration in Dome C ($3.9\,L^{-1}$) is more than an order of magnitude lower in comparison to Fairbanks ($9.0 \cdot 10^1\,L^{-1}$).".*

- *Lines 415-416 now read "This results in a higher mean $k_e$ value of 4.35 in comparison to 4.22 at Dome C and a with 43\,% much larger fraction of rough particles in comparison to 2\,% at Dome C according to the machine learning classification.".*

- *Lines 421-422 now read "The mean particle concentration during cpdd events ($9.1 \cdot 10^{-1}\,L^{-1}$) was lower than the mean of ice fog events ($3.9\,L^{-1}$).".*

*.*

*In lines 154-155 it was specified that the lidar backscattering signal is meant: "The lidar backscattering signal is proportional to the particle concentration and particle scattering cross-section.".*

*Lines 200-202 were adapted to highlight the mechanism of the summertime temperature inversions: "Strong summertime temperature inversions on the Antarctic plateau result from surface air cooling at low sun elevations. The trend looks similar to Fig. 3a, because surface air cooling increases the relative humidity due to conservation of the amount of water in the atmosphere.".*

*Local times of the day were added to the ice fog description in lines 218-219 "This is early local morning at Dome C (local solar midnight: 23:33 UTC), when nocturnal cooling has induced a high enough supersaturation for ice nucleation to occur." and lines 248-249 "This is local afternoon at Dome C (local solar noon: 03:33 UTC), when convective forcing starts to decrease again".*

*"it" in line 466 was changed to "The machine learning method" to clarify the sentence. Lines 466-468 now read "The machine learning method also detects a higher fraction of pristine particles for particle sizes larger than 40 μm in comparison to the combined fraction of columns and plates classified with the Fourier analysis.".*

**References**

Aubry, C., Delanoë, J., Groß, S., Ewald, F., Tridon, F., Jourdan, O., and Mioche, G.: Lidar–radar synergistic method to retrieve ice, supercooled water and mixed-phase cloud properties, Atmospheric Measurement Techniques, 17, 3863–3881, https://doi.org/10.5194/amt-17-3863-2024, 2024.

Bailey, M. P. and Hallett, J.: A comprehensive habit diagram for atmospheric ice crystals: Confirmation from the laboratory, AIRS II, and other field studies, Journal of the Atmospheric Sciences, 66, 2888–2899, https://doi.org/10.1175/2009jas2883.1, 2009.

Cox, C. J., Noone, D. C., Berkelhammer, M., Shupe, M. D., Neff, W. D., Miller, N. B., Walden, V. P., and Steffen, K.: Supercooled liquid fogs over the central Greenland Ice Sheet, Atmospheric Chemistry and Physics, 19, 7467–7485, https://doi.org/10.5194/acp-19-7467-2019, 2019.

Dalrymple, P. C.: A physical climatology of the Antarctic Plateau, Studies in Antarctic Meteorology, 9, 195–231, https://doi.org/10.1029/ar009p0195, 1966.

Del Guasta, M.: ICE-CAMERA: a flatbed scanner to study inland Antarctic polar precipitation, Atmospheric Measurement Techniques Discussions, 2022, 1–35, https://doi.org/10.5194/amt-15-6521-2022, 2022.

Del Guasta, M., Ricaud, P., Scarchilli, C., and Dreossi, G.: A statistical study of precipitation on the eastern antarctic plateau (Dome-C) using remote sensing and in-situ instrumentation, Polar Science, 42, 101 106, https://doi.org/10.1016/j.polar.2024.101106, 2024.

Di Natale, G., Turner, D. D., Bianchini, G., Del Guasta, M., Palchetti, L., Bracci, A., Baldini, L., Maestri, T., Cossich, W., Martinazzo, M., et al.: Consistency test of precipitating ice cloud retrieval properties obtained from

the observations of different instruments operating at Dome C (Antarctica), Atmospheric Measurement Techniques, 15, 7235–7258, https://doi.org/10.5194/amt-15-7235-2022, 2022.

Dittmann, A., Schlosser, E., Masson-Delmotte, V., Powers, J. G., Manning, K. W., Werner, M., and Fujita, K.: Precipitation regime and stable isotopes at Dome Fuji, East Antarctica, Atmospheric Chemistry and Physics, 16, 6883–6900, https://doi.org/10.5194/acp-16-6883-2016, 2016.

Genthon, C., Piard, L., Vignon, E., Madeleine, J.-B., Casado, M., and Gallée, H.: Atmospheric moisture supersaturation in the near-surface atmosphere at Dome C, Antarctic Plateau, Atmospheric Chemistry and Physics, 17, 691–704, https://doi.org/10.5194/acp-17-691-2017, 2017.

Girard, E. and Blanchet, J.-P.: Simulation of Arctic diamond dust, ice fog, and thin stratus using an explicit aerosol–cloud–radiation model, Journal of the Atmospheric Sciences, 58, 1199–1221, https://doi.org/10.1175/1520-0469(2001)058⟨1199:soaddi⟩2.0.co;2, 2001a.

Girard, E. and Blanchet, J.-P.: Microphysical parameterization of Arctic diamond dust, ice fog, and thin stratus for climate models, Journal of the Atmospheric Sciences, 58, 1181–1198, https://doi.org/10.1175/1520-0469(2001)058⟨1181:mpoadd⟩2.0.co;2, 2001b.

Gultepe, I., Zhou, B., Milbrandt, J., Bott, A., Li, Y., Heymsfield, A. J., Ferrier, B., Ware, R., Pavolonis, M., Kuhn, T., et al.: A review on ice fog measurements and modeling, Atmospheric research, 151, 2–19, 2015.

Gultepe, I., Heymsfield, A. J., Gallagher, M., Ickes, L., and Baumgardner, D.: Ice fog: The current state of knowledge and future challenges, Meteorological Monographs, 58, 4–1, https://doi.org/10.1175/amsmonographs-d-17-0002.1, 2017.

Hamel, A., Schnaiter, M., Saito, M., Wagner, R., and Järvinen, E.: Cloud Chamber Studies on the Linear Depolarisation Ratio of Small Cirrus Ice Crystals, EGUsphere, 2025, 1–32, https://doi.org/10.5194/egusphere-2025-3515, 2025.

Heumann, K. G.: Determination of inorganic and organic traces in the clean room compartment of Antarctica, Analytica chimica acta, 283, 230–245, https://doi.org/10.1016/0003-2670(93)85227-b, 1993.

Järvinen, E., Schnaiter, M., Mioche, G., Jourdan, O., Shcherbakov, V. N., Costa, A., Afchine, A., Krämer, M., Heidelberg, F., Jurkat, T., et al.: Quasi-spherical ice in convective clouds, Journal of the Atmospheric Sciences, 73, 3885–3910, https://doi.org/10.1175/jas-d-15-0365.1, 2016.

Kaye, P. H., Hirst, E., Greenaway, R. S., Ulanowski, Z., Hesse, E., DeMott, P. J., Saunders, C., and Connolly, P.: Classifying atmospheric ice crystals

by spatial light scattering, Optics letters, 33, 1545–1547, https://doi.org/10.1364/ol.33.001545, 2008.

Koop, T., Luo, B., Tsias, A., and Peter, T.: Water activity as the determinant for homogeneous ice nucleation in aqueous solutions, Nature, 406, 611–614, https://doi.org/10.1038/35020537, 2000.

Kuhn, T. and Gultepe, I.: Ice fog and light snow measurements using a high-resolution camera system, Pure and Applied Geophysics, 173, 3049–3064, 2016.

Lawson, R. P., Baker, B. A., Zmarzly, P., O'Connor, D., Mo, Q., Gayet, J.-F., and Shcherbakov, V.: Microphysical and optical properties of atmospheric ice crystals at South Pole Station, Journal of Applied Meteorology and Climatology, 45, 1505–1524, https://doi.org/10.1175/jam2421.1, 2006.

Libois, Q., Picard, G., Arnaud, L., Morin, S., and Brun, E.: Modeling the impact of snow drift on the decameter-scale variability of snow properties on the Antarctic Plateau, Journal of Geophysical Research: Atmospheres, 119, 11–662, https://doi.org/10.1002/2014jd022361, 2014.

Lu, R.-S., Tian, G.-Y., Gledhill, D., and Ward, S.: Grinding surface roughness measurement based on the co-occurrence matrix of speckle pattern texture, Applied Optics, 45, 8839–8847, https://doi.org/10.1364/ao.45.008839, 2006.

Mishra, S., Mitchell, D. L., Turner, D. D., and Lawson, R.: Parameterization of ice fall speeds in midlatitude cirrus: Results from SPartICus, Journal of Geophysical Research: Atmospheres, 119, 3857–3876, 2014.

Palm, S. P., Yang, Y., Spinhirne, J. D., and Marshak, A.: Satellite remote sensing of blowing snow properties over Antarctica, Journal of Geophysical Research: Atmospheres, 116, https://doi.org/10.1029/2011jd015828, 2011.

Pomeroy, J.: A process-based model of snow drifting, Annals of Glaciology, 13, 237–240, https://doi.org/10.3189/s0260305500007965, 1989.

Ricaud, P., Bazile, E., del Guasta, M., Lanconelli, C., Grigioni, P., and Mahjoub, A.: Genesis of diamond dust, ice fog and thick cloud episodes observed and modelled above dome c, Antarctica, Atmospheric Chemistry and Physics, 17, 5221–5237, https://doi.org/10.5194/acp-17-5221-2017, 2017.

Santachiara, G., Belosi, F., and Prodi, F.: Ice crystal precipitation at Dome C site (East Antarctica), Atmospheric Research, 167, 108–117, https://doi.org/10.1016/j.atmosres.2015.08.006, 2016.

Schmitt, C. G., Stuefer, M., Heymsfield, A. J., and Kim, C. K.: The microphysical properties of ice fog measured in urban environments of Interior Alaska, Journal of Geophysical Research: Atmospheres, 118, 11–136, https://doi.org/10.1002/jgrd.50822, 2013.

Schmitt, C. G., Järvinen, E., Schnaiter, M., Vas, D., Hartl, L., Wong, T., and Stuefer, M.: Classification of ice particle shapes using machine learning on forward light scattering images, Artificial Intelligence for the Earth Systems, https://doi.org/10.1175/aies-d-23-0091.1, 2024a.

Schmitt, C. G., Vas, D., Schnaiter, M., Järvinen, E., Hartl, L., Wong, T., Cassella, V., and Stuefer, M.: Microphysical characterization of boundary layer ice particles: results from a 3-year measurement campaign in interior Alaska, Journal of Applied Meteorology and Climatology, https://doi.org/10.1175/jamc-d-23-0190.1, 2024b.

Schnaiter, M., Järvinen, E., Vochezer, P., Abdelmonem, A., Wagner, R., Jourdan, O., Mioche, G., Shcherbakov, V., Schmitt, C., Tricoli, U., et al.: Cloud chamber experiments on the origin of ice crystal complexity in cirrus clouds, Atmospheric Chemistry and Physics, 16, 5091–5110, https://doi.org/10.5194/acp-16-5091-2016, 2016.

Schneider, J., Höhler, K., Wagner, R., Saathoff, H., Schnaiter, M., Schorr, T., Steinke, I., Benz, S., Baumgartner, M., Rolf, C., et al.: High homogeneous freezing onsets of sulfuric acid aerosol at cirrus temperatures, Atmospheric Chemistry and Physics, 21, 14403–14425, https://doi.org/10.5194/acp-21-14403-2021, 2021.

Shupe, M. D. and Intrieri, J. M.: Cloud radiative forcing of the Arctic surface: The influence of cloud properties, surface albedo, and solar zenith angle, Journal of climate, 17, 616–628, 2004.

Sourdeval, O., Gryspeerdt, E., Krämer, M., Goren, T., Delanoë, J., Afchine, A., Hemmer, F., and Quaas, J.: Ice crystal number concentration estimates from lidar–radar satellite remote sensing–Part 1: Method and evaluation, Atmospheric Chemistry and Physics, 18, 14327–14350, https://doi.org/10.5194/acp-18-14327-2018, 2018.

Ulanowski, Z., Kaye, P. H., Hirst, E., and Greenaway, R.: Light scattering by ice particles in the Earth's atmosphere and related laboratory measurements, in: Procs 12th Int Conf on Electromagnetic and Light Scattering, University of Helsinki, URL https://api.semanticscholar.org/CorpusID:17077450, 2010.

Ulanowski, Z., Kaye, P. H., Hirst, E., Greenaway, R., Cotton, R. J., Hesse, E., and Collier, C. T.: Incidence of rough and irregular atmospheric ice particles from Small Ice Detector 3 measurements, Atmospheric Chemistry and Physics, 14, 1649–1662, https://doi.org/10.5194/acp-14-1649-2014, 2014.

Vanderpool, R. W., Krug, J. D., Kaushik, S., Gilberry, J., Dart, A., and Witherspoon, C. L.: Size-selective sampling performance of six low-volume "total" suspended particulate (TSP) inlets, Aerosol Science and Technology, 52, 98–113, https://doi.org/10.1080/02786826.2017.1386766, 2018.

Vignon, É., Raillard, L., Genthon, C., Del Guasta, M., Heymsfield, A. J., Madeleine, J.-B., and Berne, A.: Ice fog observed at cirrus temperatures at Dome C, Antarctic Plateau, Atmospheric Chemistry and Physics, 22, 12 857–12 872, https://doi.org/10.5194/acp-22-12857-2022, 2022.

Virkkula, A., Grythe, H., Backman, J., Petäjä, T., Busetto, M., Lanconelli, C., Lupi, A., Becagli, S., Traversi, R., Severi, M., et al.: Aerosol optical properties calculated from size distributions, filter samples and absorption photometer data at Dome C, Antarctica, and their relationships with seasonal cycles of sources, Atmospheric Chemistry and Physics, 22, 5033–5069, https://doi.org/10.5194/acp-22-5033-2022, 2022.

Vochezer, P., Järvinen, E., Wagner, R., Kupiszewski, P., Leisner, T., and Schnaiter, M.: In situ characterization of mixed phase clouds using the Small Ice Detector and the Particle Phase Discriminator, Atmospheric Measurement Techniques, 9, 159–177, https://doi.org/10.5194/amt-9-159-2016, 2016.

Walden, V. P., Warren, S. G., and Tuttle, E.: Atmospheric ice crystals over the Antarctic Plateau in winter, Journal of Applied Meteorology, 42, 1391–1405, https://doi.org/10.1175/1520-0450(2003)042⟨1391:aicota⟩2.0.co;2, 2003.

Zeng, X.: Radiatively induced precipitation formation in diamond dust, Journal of Advances in Modeling Earth Systems, 10, 2300–2317, https://doi.org/10.1029/2018ms001382, 2018.